# Microtubule-dependent and independent roles of spastin in lipid droplet dispersion and biogenesis

Nimesha Tadepalle[1,2], Lennart Robers[1,2], Matteo Veronese[1,2], Peter Zentis[2] , Felix Babatz[2], Susanne Brodesser[2] , Anja V Gruszczyk[1,2], Astrid Schauss[2], Stefan Höning[3], Elena I Rugarli[1,2] 

**Lipid droplets (LDs) are metabolic organelles that store neutral lipids and dynamically respond to changes in energy availability by accumulating or mobilizing triacylglycerols (TAGs). How the plastic behavior of LDs is regulated is poorly understood. Hereditary spastic paraplegia is a central motor axonopathy predominantly caused by mutations in *SPAST*, encoding the microtubule-severing protein spastin. The spastin-M1 isoform localizes to nascent LDs in mammalian cells; however, the mechanistic significance of this targeting is not fully explained. Here, we show that tightly controlled levels of spastin-M1 are required to inhibit LD biogenesis and TAG accumulation. Spastin-M1 maintains the morphogenesis of the ER when TAG synthesis is prevented, independent from microtubule binding. Moreover, spastin plays a microtubule-dependent role in mediating the dispersion of LDs from the ER upon glucose starvation. Our results reveal a dual role of spastin to shape ER tubules and to regulate LD movement along microtubules, opening new perspectives for the pathogenesis of hereditary spastic paraplegia.**

## Introduction

Lipid droplets (LDs) are cytosolic organelles specialized in storing neutral lipids such as triacylglycerols (TAGs) and cholesterol esters. LDs segregate a lipid core within a phospholipid monolayer coated with specific proteins and play crucial roles by protecting cells from lipotoxicity, serving as energy reservoirs, forming compartments for protein storage or degradation, and mediating lipid trafficking between organelles and membranes (Listenberger et al, 2003; Cermelli et al, 2006; Zehmer et al, 2009; Moldavski et al, 2015; Schuldiner & Bohnert, 2017). LD dysfunction has been implicated in several diseases, including obesity, cancer, and surprisingly also neurodegeneration (Krahmer et al, 2013; Welte, 2015; Walther et al, 2017; Pennetta & Welte, 2018).

Biogenesis of LDs occurs de novo from the tubular ER and follows distinct steps, beginning with the accumulation of neutral lipids between the two leaflets of the ER membrane to form a lens-like structure that progressively grows into nascent LDs that then expand into mature LDs (Pol et al, 2014; Walther et al, 2017). Remarkably, LDs maintain a unique intimate relationship with the ER. In yeast, LDs remain connected to the ER membrane (Jacquier et al, 2011), and membrane bridges between the ER and LDs have been revealed by various imaging techniques in eukaryotic cells (Kassan et al, 2013; Wilfling et al, 2013; Valm et al, 2017). Connections between the ER and the LDs are crucial to allow some enzymes involved in the last step of TAG synthesis to relocalize from the ER to LDs (Wilfling et al, 2013). The exact mechanism that permits LDs to bud into the cytoplasm is still debated, and it is unknown if specific molecular components regulate this process (Walther et al, 2017). LDs interact also with other organelles such as mitochondria, peroxisomes, and lysosomes (Schuldiner & Bohnert, 2017).

The ER is a complex and dynamic structure that undergoes continuous rearrangements, including tubule elongation, retraction, and fusion (Voeltz et al, 2006; Westrate et al, 2015). This plasticity depends on the cytoskeleton and especially on microtubules (MTs) with which the ER network maintains strict association (Gurel et al, 2014). MTs have also been involved in several aspects of LD behavior, including movement, fusion, and turnover (Gross et al, 2000; Bostrom et al, 2005; Shubeita et al, 2008; Welte, 2009; Orlicky et al, 2013; Herms et al, 2015).

Severing of MTs represents a powerful way to rearrange the cytoskeleton, by producing short and dynamic MTs and thereby increasing MT mass (Sharp & Ross, 2012). MT severing is catalyzed by enzymes that use the power generated by ATP hydrolysis to introduce breaks into the MT lattice (Roll-Mecak & McNally, 2010). Among them, spastin is encoded by the *SPAST* gene, which is mutated in the most common form of dominant hereditary spastic paraplegia (HSP), a progressive neurological disease characterized by the dying-back of the long corticospinal axons (Hazan et al, 1999; Errico et al, 2002; Evans et al, 2005; Roll-Mecak & Vale, 2005, 2008; Reid & Rugarli, 2010; Fink, 2014). Spastin has been implicated in

[1]Institute for Genetics, University of Cologne, Cologne, Germany  [2]Cologne Excellence Cluster on Cellular Stress Responses in Aging-Associated Diseases (CECAD), Cologne, Germany  [3]Institute for Biochemistry I, University of Cologne, Cologne, Germany

Correspondence: elena.rugarli@uni-koeln.de
Nimesha Tadepalle's present address is Molecular and Cell Biology Laboratory, Salk Institute of Biological Sciences, La Jolla, CA, USA

various processes characterized by MT rearrangements, such as axonal branching and neurite formation (Yu et al, 2008; Brill et al, 2016), synaptic function (Sherwood et al, 2004; Trotta et al, 2004; Riano et al, 2009), axonal regeneration (Stone et al, 2012), endosome tubulation (Allison et al, 2013), nuclear envelope breakdown (Vietri et al, 2015), progression of mitosis (Zhang et al, 2007), and midbody abscission (Connell et al, 2009).

Spastin is synthesized in two isoforms, owing to alternative initiation of translation (Claudiani et al, 2005). Whereas the shorter and more abundant spastin-M87 isoform localizes mainly to the cytosol and endosomal compartments, the longer spastin-M1 isoform is bound to the ER (Connell et al, 2009; Park et al, 2010). Transcriptional and translational mechanisms ensure that the levels of spastin-M1 are kept significantly lower than those of spastin-M87 (Claudiani et al, 2005; Schickel et al, 2007; Mancuso & Rugarli, 2008), suggesting that overexpression of this isoform may be toxic. When cells are loaded with oleic acid (OA) and accumulate LDs, spastin-M1 is targeted to LDs (Papadopoulos et al, 2015; Chang et al, 2019). Spastin-M1 has a topology similar to other LD proteins, as it contains a rather short hydrophobic region interrupted by a positively charged residue that forms a hairpin in the ER membrane and allows its mobilization to the LD phospholipid monolayer (Park et al, 2010; Papadopoulos et al, 2015; Chang et al, 2019). Recently, a role of spastin-M1 in tethering LDs to peroxisomes for trafficking of fatty acids has been shown in human cells (Chang et al, 2019). Furthermore, manipulation of spastin levels in invertebrate organisms leads to tissue-specific phenotypes characterized by abnormalities in LD size and number (Papadopoulos et al, 2015), raising the question if spastin-M1 also regulates LD biogenesis. Understanding the functions of spastin-M1 is crucial because this isoform is highly expressed in the brain and specifically interacts with other HSP proteins, such as atlastin1 and REEP1 (Errico et al, 2004; Solowska et al, 2008; Blackstone, 2018), indicating that it may play a fundamental role in the pathogenesis of the disease.

Here, we show that lack of spastin in murine cell lines leads to increased LD biogenesis and accumulation of TAGs. This phenotype results from both MT-dependent and MT-independent functions of spastin-M1. On the one hand, increased LD biogenesis buffers the loss of spastin-M1 at the ER, independently from the ability of spastin to bind the MTs. On the other hand, lack of spastin-mediated MT-severing causes LD clustering and failure to disperse LD upon glucose deprivation. Notably, the levels of spastin-M1 are crucial to maintain LD homeostasis because both overexpression and loss of spastin-M1 result in similar phenotypes. Our data reveal a novel link between spastin-M1 and LD biogenesis and distribution and open new perspectives for the pathogenesis of HSP.

# Results

## Spastin KO in immortalized motoneurons leads to accumulation of LDs and TAGs

To explore the molecular role of spastin in LD biology in mammalian cells, we used CRISPR-Cas9 gene editing to disrupt the *Spast* gene in NSC34 cells. These cells are murine-immortalized motoneurons that express high levels of spastin-M1 (Cashman et al, 1992;

Errico et al, 2004). Moreover, upon OA addition, spastin-M1 is recovered in the LD fraction in NCS34 cells (Papadopoulos et al, 2015). We targeted exon 5 of the *Spast* gene with two specific gRNAs to induce an out-of-frame deletion and abolish gene function (Fig S1A). We obtained one clone that showed complete absence of the spastin protein by both Western blot and immunofluorescence analysis (Fig S1B and C). Quantitative analysis of the *Spast* transcript levels showed a significant down-regulation in the KO cells, suggestive of nonsense-mediated decay (Fig S1D). Subcloning and sequencing of the targeted genomic region revealed six different targeted alleles carrying disrupting deletions in *Spast* exon 5, in agreement with the polyploidy of the cells (Fig S1E).

We then asked if cells lacking spastin showed any difference in LD content. PLIN2 is a LD-targeted protein commonly used as a marker for these organelles (Bickel et al, 2009). Western blot analysis of PLIN2 revealed increased levels in KO compared with wild-type (WT) cells under basal conditions (Fig 1A and B). To ensure that these results reflect an increase in LD content, we investigated the localization of PLIN2 by immunofluorescence. Under basal conditions, the PLIN2 signal was diffused in the cytosol and only occasionally labelled ring-like structures reminiscent of LDs (Fig 1C), in agreement with scarcity of LDs in neuron-like cells. However, PLIN2-positive structures were more evident in KO cells (Fig 1C). We quantified major TAG species carrying C16:0, C18:0, C18:1, and C18:2 fatty acyl chains by tandem mass spectrometry in WT and KO cells. These experiments showed that TAGs were increased in KO compared with WT cells, largely independent from the saturation and the length of the acyl chain (Figs 1D and S2).

We then assessed LD content under starvation conditions (Fig 1A and B). Mild starvation such as glucose or serum deprivation can affect LD content by inducing lipophagy (Singh & Cuervo, 2012), whereas strong starvation using HBSS (combining serum amino acids, and glucose starvation) stimulates LD biogenesis by mobilizing membrane lipids via autophagy (Rambold et al, 2015). Spastin KO cells displayed increased PLIN2 levels also in conditions of serum and HBSS starvation (Fig 1A and B). After overnight incubation in HBSS, the number and the area occupied by PLIN2-positive structures and the levels of TAGs were increased in the KO compared with WT cells (Figs 1E–H and S2).

In conclusion, lack of spastin in NSC34 cells leads to increased LD and TAG content both in basal and starved conditions.

## Spastin reexpression rescues LD and TAG accumulation

Spastin synthesis is regulated by two alternative start codons present in the first exon of the *Spast* mRNA. The first start codon is embedded in a poor Kozak consensus sequence (tgaATGA, here called endogenous Kozak), leading to reinitiation of translation at a downstream ATG (Claudiani et al, 2005). As a result, spastin-M1 is produced at lower levels, whereas spastin-M87 represents the major isoform expressed in cells and tissues. Because spastin-M1 resides at the ER and can localize to LDs (Park et al, 2010; Papadopoulos et al, 2015), we asked if lack of this isoform is exquisitely responsible for the LD phenotype of spastin KO cells. To this purpose, we produced retroviral vectors expressing different human *SPAST* cDNAs and infected KO cells to restore spastin expression (Fig 2A). Spastin expression levels and relative phenotypes were evaluated on a mixed cell population.

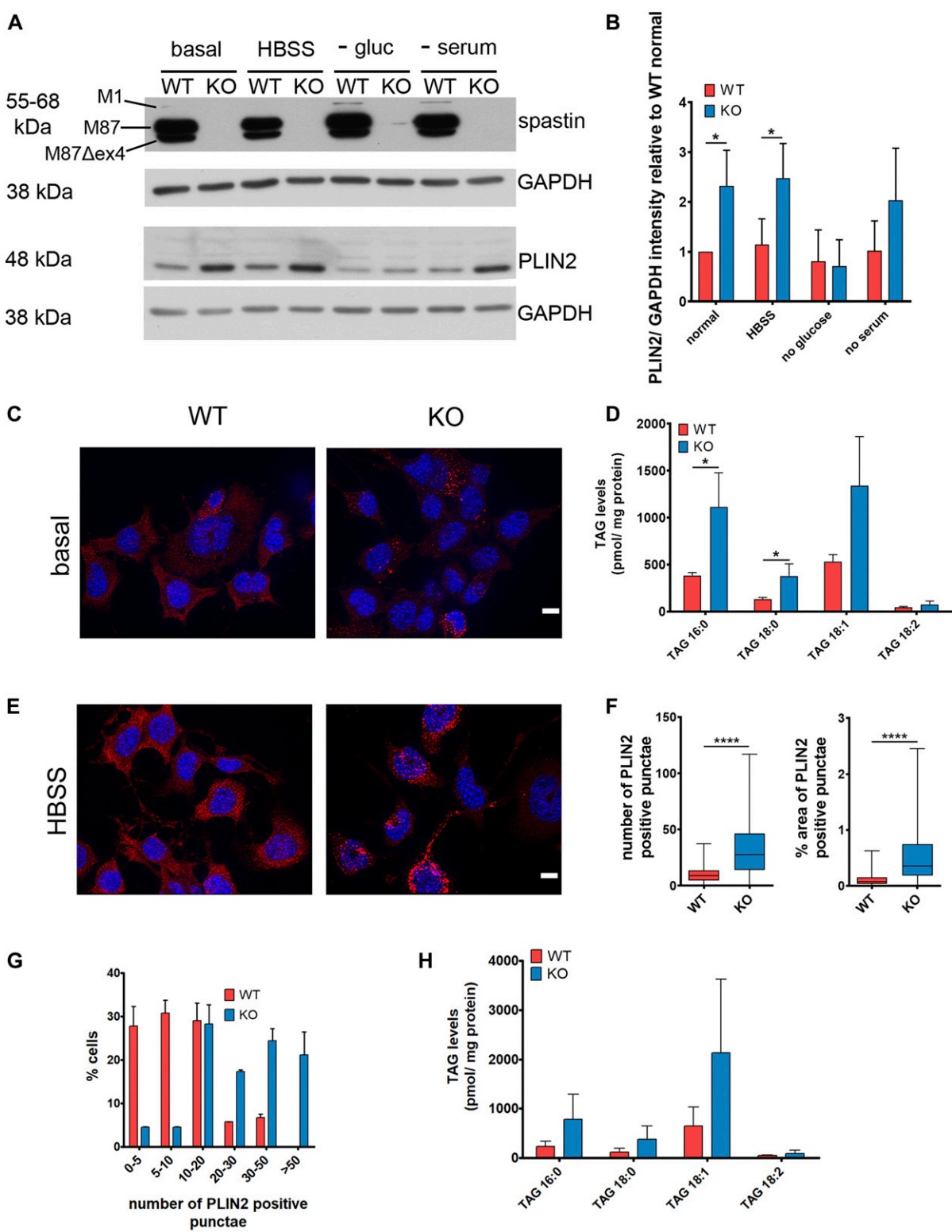

**Figure 1. *Spast* deletion increases lipid droplets and TAG content in immortalized mouse motoneurons.**
**(A)** Western blot analysis of PLIN2 levels in WT and spastin KO NSC34 cells cultured in basal medium or starved overnight as indicated. **(B)** Quantification of PLIN2 levels normalized to GAPDH and relative to WT under basal conditions. Bars show mean ± SD. Unpaired *t* test *$P$ < 0.05 (n = 3). **(C)** Immunofluorescence analysis of endogenous PLIN2 levels in WT and KO cells under basal conditions. Representative maximum projection images are shown. Scale bar: 10 $\mu$m. **(D)** TAG content of WT and KO cells under basal conditions (without oleic acid). **(E)** Same as (C) under HBSS starvation conditions. **(F)** Box and whisker plots of the number and % of area occupied by PLIN2-positive punctae in WT and KO cells in HBSS (≈300 cells/genotype from three experiments). Unpaired *t* test ****$P$ < 0.0001. **(G)** Distribution of cells with various ranges of PLIN2-positive punctae under HBSS starvation conditions (n = 3, ≈300 cells/genotype; $\chi^2$ test $P$ < 0.001). **(H)** TAG content of WT and KO cells under HBSS starvation conditions. Bars represent means ± SD of three independent biological replicates. See also Figs S1 and S2.

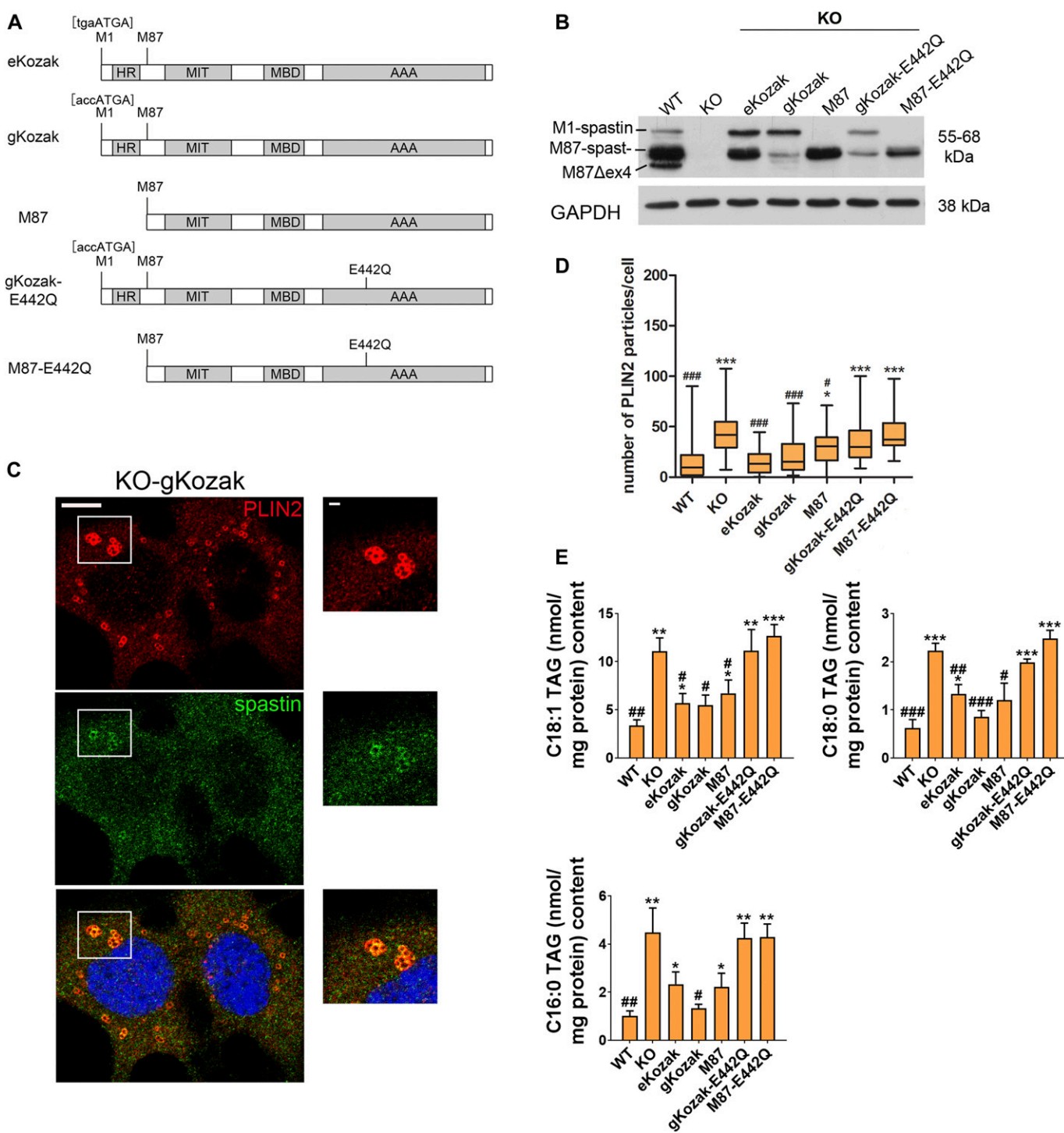

**Figure 2. Spastin reexpression rescues lipid droplet and TAG accumulation.**
**(A)** Schematic representation of the rescue constructs expressing human spastin. The positions of M1 and M87 and the Kozak sequences around the first start codon are shown. The Kozak sequence surrounding the second start codon has not been changed. **(B)** Representative Western blot showing spastin-M1 and M87 levels in KO cells infected with the indicated viral construct. **(C)** Single-plane confocal images of KO-gKozak cell line treated with oleic acid and stained for spastin (green) and PLIN2 (red). Scale bar: 10 $\mu$m. Enlargements of boxed area are shown on the right (scale bar: 1 $\mu$m). **(D)** Box plot of PLIN2 particles per cell (n = 3 independent experiments, ≈200 cells/genotype). One-way ANOVA with post-Tukey test: *; #P < 0.05, ***, ###P < 0.001. * refers to P versus WT, whereas # refers to P versus KO. **(E)** TAG content of different rescue cell lines upon HBSS starvation. Unpaired t test *; # < 0.05, **, ##P < 0.01. ***, ###P < 0.001. See also Fig S3. AAA, ATPase domain; HR, hairpin domain; MBD, microtubule-binding domain; MIT, microtubule interacting and trafficking domain.

As predicted, a retroviral vector containing the human *SPAST* open reading frame with the first start codon surrounded by the endogenous Kozak (eKozak) sequence expressed lower levels of spastin-M1 and higher amounts of spastin-M87 (Fig 2A and B). By immunofluorescence, spastin staining was similar to that observed in WT cells (Fig S3). In contrast, when the vector was modified to encode a good Kozak sequence (accATGA, here dubbed as gKozak), spastin-M1 was preferentially synthesized (Fig 2A and B). By immunofluorescence, spastin produced by this vector localized to very small punctae in the cytoplasm and decorated the surface of LDs when cells were loaded with OA (Figs 2C and S3). Finally, the M87 retroviral vector contained a truncated *SPAST* cDNA that efficiently expressed only spastin-M87 (using the endogenous Kozak sequence) at significant levels and showed a diffuse cytosolic localization (Figs 2A and B and S3). It should be noted that these constructs express only forms of spastin containing exon 4 and, therefore, do not fully reestablish endogenous spastin levels. So far, it is not known whether Δex4-spastin alternative isoforms play any specific functional role.

To monitor rescue of the phenotype, we examined the number of PLIN2-labeled puncta by immunofluorescence and measured TAGs by mass spectrometry. Remarkably, both constructs expressing spastin-M1 (eKozak and gKozak) significantly decreased the number of PLIN2-stained structures and reduced TAG levels in KO cells. Expression of spastin-M87 alone partially rescued TAG accumulation and reduced PLIN2 accumulation (Fig 2D and E). Expression of mutant forms of spastin-M1 and/or spastin-M87 carrying the E442Q substitution in the Walker B motif of the AAA domain (gKozak-E442Q and M87-E442Q; Fig 2A and B), which abolishes ATPase activity, did not reduce the number of PLIN2-positive punctae and the TAG accumulation in KO cells (Fig 2D and E). However, these mutant forms were toxic and could not be expressed at the same levels as wild-type spastin, preventing to conclusively assess the role of a functional AAA domain. These results provide a causal relationship between lack of spastin and the perturbation of lipid metabolism and implicate spastin-M1 and, to a lesser extent, spastin-M87 in the phenotype.

### Spastin limits TAG synthesis independently from the source of fatty acids

Accumulation of LDs in the absence of spastin could result from increased synthesis or decreased utilization of neutral lipids. To assess TAG synthesis, we fed WT and KO NSC34 cells with $^{14}$C-labeled OA for different time points and monitored the incorporation of the radiolabeled OA into TAG using TLC followed by autoradiography. These experiments revealed an accelerated incorporation of $^{14}$C-OA in TAGs in KO compared with WT cells (Fig 3A and B). To examine lipolysis, cells were loaded with $^{14}$C-OA for 24 h, followed by treatment with triacsin C, an inhibitor of long-chain fatty acyl-CoA synthetases (Kim et al, 2001), for additional 6 h to block TAG synthesis. TLC was then performed to detect the levels of intracellular TAGs and the release of OA into the medium. As expected, KO cells had incorporated more $^{14}$C-OA in TAGs than WT cells after the 24-h incubation; however, the rate of TAG utilization was comparable with that of WT cells during the 6-h incubation with triacsin C (Fig 3C–E). Overall, these experiments indicate that when

challenged with exogenous OA, spastin KO cells form more LDs owing to increased synthesis of TAGs.

To assess lipolysis also under basal conditions, we examined if down-regulating the main lipase, adipose triglyceride lipase (ATGL), affects LD content differently in WT and KO cells. ATGL was increased in NSC34 KO cells (Fig 3F), probably as a compensatory mechanism. As expected, down-regulation of *Atgl* increased PLIN2 levels both in WT and KO cells, but did not abrogate the difference between the two genotypes, making it unlikely that impaired lipolysis contributes to LD accumulation (Fig 3F and G). We then tested the contribution of lipophagy to LD content under basal conditions (Singh & Cuervo, 2012). To this end, we incubated WT and KO cells with chloroquine which blocks lysosomal degradation. Surprisingly, we found that the number of PLIN2 particles significantly decreased both in WT and KO cells; however, the number of PLIN2 particles remained higher in KO than in WT cells also after chloroquine treatment (Fig S4A).

The previous data indicate that in NSC34 cells, autophagy plays a constitutive role in LD formation, rather than in LD consumption. Autophagy has been previously shown to mediate increase in LD biogenesis upon HBSS starvation (Rambold et al, 2015), so we hypothesized that even in the absence of spastin cells, fatty acids released by autophagy in LDs are stored. Consistently, spastin KO cells displayed an intact autophagic flux, monitored by p62 accumulation and conversion of LC3-I to LC3-II upon bafilomycin inhibition both in the basal and HBSS conditions (Fig S4B). To directly assess the conversion from autophagosomes to autolysosomes, we transfected cells with a construct expressing mCherry-GFP-LC3B (Pankiv et al, 2007). Because the GFP signal is quenched by the acidic pH of lysosomes, this construct allows following not only the formation of autophagosomes but also their fusion with lysosomes. Under basal conditions, KO cells showed an increased number of autophagosomes and autolysosomes compared with WT cells (Fig S4C and D). Stimulation of autophagy by 4 h of incubation in HBSS led to an increase in the number of autophagosomes in both WT and KO cells compared with basal conditions; however, in WT cells autolysosomes were also increased, whereas the latter were not significantly different from basal conditions in KO cells (Fig S4C and E). These data suggest an increase in the autophagic flux in the absence of spastin, which is also supported by decreased levels of the autophagic adaptor p62 (Fig S4F and G). Thus, enhanced autophagy may contribute to deliver fatty acids for TAG synthesis in KO cells. Incubation of cells with HBSS and 3-methyladenine or bafilomycin A, which inhibit autophagy either at early or late steps, respectively, was, however, not sufficient to abrogate the increased levels of LDs (visualized via PLIN2 levels) in KO compared with WT cells (Fig S4F and H). Taken together, our data indicate that LD biogenesis is enhanced in cells lacking spastin, both when fatty acids are supplied exogenously and when derive from autophagic degradation.

### Increased LD biogenesis buffers lack spastin-M1 at the ER

To dissect the mechanism leading to LD accumulation in the absence of spastin, we sought a different cellular system allowing to better image LDs and their relation with the ER. To this end, we derived MEFs from a *Spast* KO mouse model, which carries a T-to-G

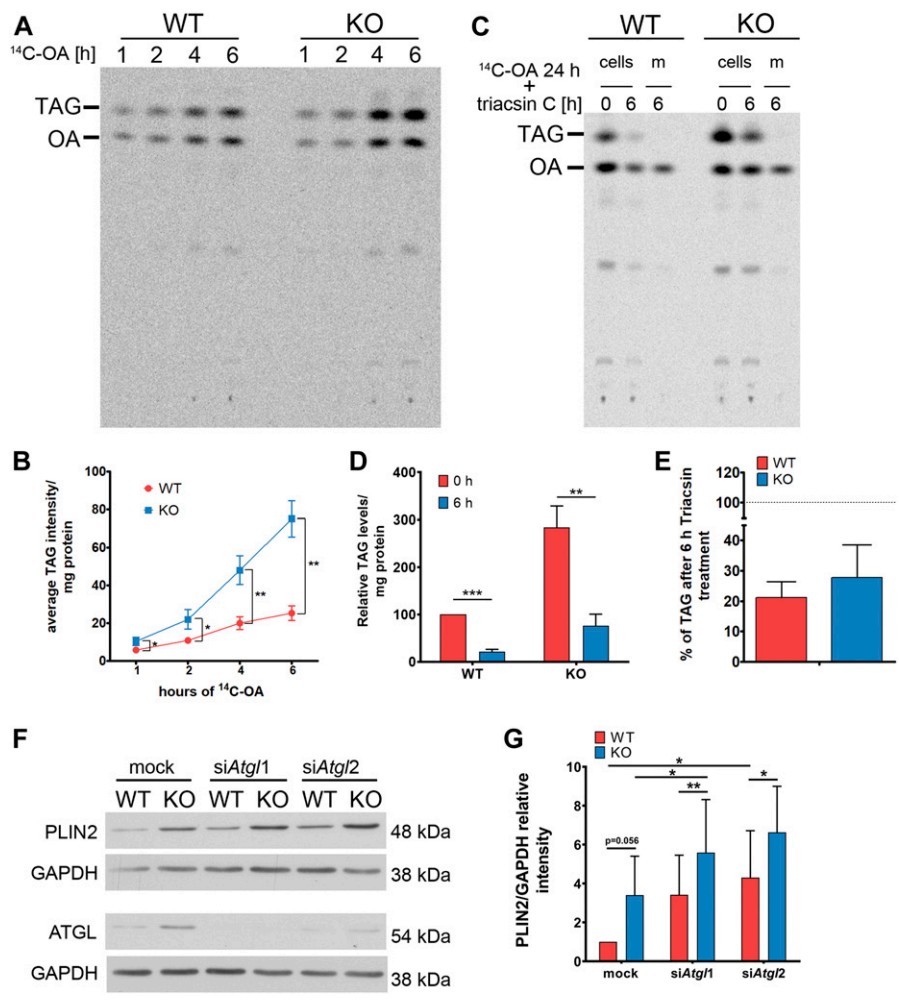

**Figure 3.  Lack of spastin increases TAG synthesis.**
**(A)** Representative autoradiogram of TLC plate of WT and KO NSC34 cells treated with $^{14}$C-OA for the indicated time points. **(B)** Quantification of TAG accumulation over time. n = 3, unpaired $t$ test, $*P < 0.05$, $**P < 0.01$. **(C)** Representative autoradiogram of TLC plates showing TAG levels of WT and KO cells or media (m) after 24-h treatment with $^{14}$C-OA (0 h) followed by 6-h incubation with triacsin C (6 h). **(D)** Quantification of the TAG content of experiments shown in (C), normalized to WT cells at 0 h. n = 3, unpaired $t$ test, $**P < 0.01$, $***P < 0.001$. **(E)** Quantification of the decrease in TAG content after 6 h of triacsin C treatment in WT and KO cells (dotted line represents levels at the beginning of treatment, 0 h, of the respective genotype). **(F)** WT and spastin KO NSC34 cells were either mock treated or down-regulated with two different siRNAs for *Atgl* (si*Atgl*1 and si*Atgl*2) and PLIN2 levels were analyzed by Western blot. **(G)** Quantification of PLIN2 level normalized to GAPDH are expressed relative to WT mock sample (n = 5, paired $t$ test $*P < 0.05$, $**P < 0.01$). See also Fig S4.

mutation in the exon 7 splice donor site (Kasher et al, 2009). The levels of PLIN2 were increased in *Spast* KO MEFs in all culture conditions, including glucose deprivation (Fig 4A and B). Increased TAGs were observed in the absence of spastin both in basal and HBSS conditions by lipidomics (Fig 4C and D). The number of LDs stained by BODIPY 493/503 was increased in KO cells upon HBSS incubation (Fig 4E). Moreover, feeding the cells with BODIPY C16 that accumulates in LDs revealed larger LDs in KO cells upon HBSS starvation (Fig S5A and B). Finally, we performed super-resolution microscopy after labeling LDs with PLIN2 antibodies to differentiate between true large LDs and aggregated smaller LDs. This showed that the LD signal in the KO cells often corresponded to clustered small LDs, but individual large LDs were also observed (Fig 4F). Thus, an independent cellular system lacking spastin recapitulates the LD accumulation.

We then examined the ER compartments where LDs are formed. These regions of the ER have been shown to persist upon serum starvation and have been named preexisting LDs (pre-LDs) (Kassan et al, 2013). Pre-LDs can be revealed before they are detected by neutral lipid-binding dyes by transfection either of the model peptide GFP-HPos or of a construct expressing acyl-CoA synthetase long-chain family member 3 (ACSL3) (Kassan et al, 2013). Previously,

we showed that spastin-M1 co-localizes with pre-LDs (Papadopoulos et al, 2015), suggesting a role of spastin in shaping these domains. We transfected WT and KO MEFs with GFP-HPos or ACSL3-GFP, serum-starved cells for 24 h, and then monitored the number of pre-LDs. Spastin KO cells showed a higher number of pre-LDs after starvation compared with WT cells (Fig 4G–I). Loading OA for 15 min increased the number of HPos or ACSL3-positive LDs in cells of both genotypes, with the number of LDs remaining higher in KO cells at the end of the treatment (Fig 4G–I). These data indicate that lack of spastin is permissive to LD formation at the ER.

Spastin-M1 may directly affect the shaping of the ER tubules by affecting the curvature via the hairpin domain, by recruiting other ER morphogens, and by interacting with the cytoskeleton via the MT-binding domain (MBD). This notwithstanding, ER morphology in spastin KO cells even in HBSS medium was indistinguishable from that of WT cells, showing perinuclear sheets and peripheral tubules (Fig 5A). We hypothesized that increased LD biogenesis, by expanding the surface of the cytosolic phospholipid leaflet of the ER membrane, may compensate for the loss of spastin-M1. We, therefore, starved MEFs with HBSS and tested the effect of inhibitors of diacylglycerol O-acyltransferase 1 and 2 (DGAT1 and DGAT2), the enzymes that convert diacylglycerol and fatty acyl-CoA to TAG,

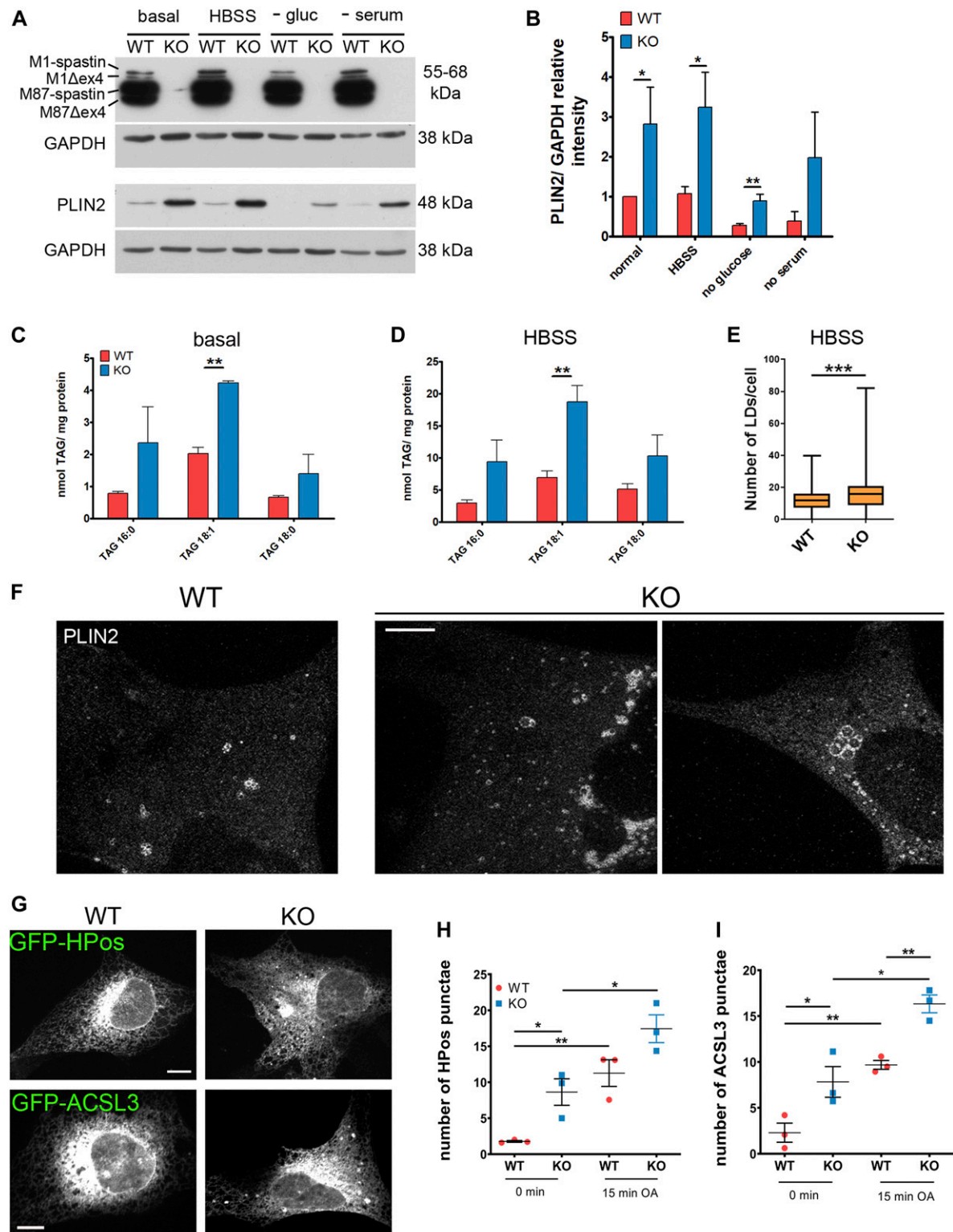

**Figure 4.** *Spast* **deletion is associated with increased pre-lipid droplets (LDs).**
**(A)** WT and spastin KO MEFs were cultured with basal medium or starved overnight as indicated and PLIN2 levels were analyzed by Western blot. **(B)** PLIN2 levels were normalized to GAPDH and are shown relative to the WT under basal conditions (n = 3, unpaired *t* test *P < 0.05, **P < 0.01). **(C, D)** TAG content of WT and KO cells under basal (C) and HBSS starvation conditions (D). Bars represent means ± SD of three independent biological replicates. Paired *t* test **P < 0.01. **(E)** MEFs were fed with BODIPY to trace LDs and treated with HBSS overnight. Data represent box plot of number of LDs/cell under HBSS starvation conditions of three independent biological replicates (≈300 cells/genotype). Unpaired *t* test ***P < 0.001. **(F)** Representative images of super-resolution single plane confocal images of WT and KO MEFs treated with HBSS and

on the tubular morphology of the ER. Intriguingly, spastin KO MEFs were vulnerable to inhibition of TAG synthesis upon starvation and showed a dramatic collapse of the ER into discrete aggregates (Fig 5A and C), suggesting that increased LD biogenesis is beneficial in the absence of spastin.

To distinguish if this phenotype is caused by the lack of a shaping function of spastin-M1 or by its interaction with the MTs, we reconstituted human wild-type or ΔMBD spastin using retroviral transduction in KO MEFs (Fig S5C). Spastin binding to the MTs is required to mediate efficient severing (Connell et al, 2009). In contrast to the E442Q mutant forms of spastin, transduction of ΔMBD variants were not toxic. We selected for further analysis clones expressing spastin-M1 and M87 at different levels and ratios: clone M1M87 expresses the two isoforms at a nearly endogenous ratio; however, the levels of both isoforms are lower than in wild-type conditions; clone M1highM87 expresses higher than endogenous levels of spastin M1 and clone M1M87low synthesizes nearly endogenous levels of spastin-M1, but very low levels of spastin-M87 (Fig 5B). We also analyzed two clones expressing similar levels of spastin-M87-ΔMBD but differing for the presence or absence of spastin-M1-ΔMBD (Fig 5B). Notably, expression of high levels of spastin M1-ΔMBD but not M87-ΔMBD in KO cells was sufficient to rescue the susceptibility to DGAT inhibition (Fig 5A–C), indicating that restoring spastin-M1 at the ER independent of its MT-binding domain is sufficient to rescue the phenotype. Surprisingly, expression of MT-severing competent spastin-M1 did not rescue the ER collapse phenotype (Fig 5A–C) likely because spastin-M1 levels were insufficient (in clone M1M87) or too high (in M1highM87) (Fig 5A–C). These data uncouple MT-dependent and ER-shaping functions of spastin-M1 and indicate that excessive MT-severing close to the ER may even be detrimental, consistent with the notion that several mechanisms are in place to restrict the expression of spastin-M1 (Claudiani et al, 2005; Mancuso & Rugarli, 2008).

### Spastin regulates dispersion of LDs upon glucose deprivation

A previous study showed that glucose deprivation stimulates LD movement from the perinuclear area to the mitochondria in Vero cells. This LD dispersion is associated with an increase in detyrosinated (ΔTyr) MTs (Herms et al, 2015). In MEFs, 2 h of glucose deprivation after OA loading was sufficient to induce an increase in ΔTyr-tubulin in both WT and KO cells, evident both in the total lysate and in the insoluble extracts, which reflect polymerized MTs (Fig S6A and B), thus confirming previous findings (Herms et al, 2015). Because spastin preferentially severs ΔTyr-MTs (Roll-Mecak & Vale, 2008), we investigated if spastin-M1 on the LDs plays any role in mediating LD movement under these conditions. To address this possibility, we loaded MEFs with OA for 24 h and then deprived them of OA and glucose for 16 h (Herms et al, 2015). OA loading induced the formation of large LDs, clustered in the perinuclear region (Fig

6A). After 16 h of glucose deprivation, LDs appeared small and mostly dispersed in the cytosol in most WT cells, independently from the extent of OA loading (Figs 6A and B and S6C and D). In contrast, a high percentage of KO cells still showed a clustered LD phenotype (Fig 6A and B). To assess LD movement, time-lapse movies were taken at the end of the glucose deprivation (Videos 1 and 2). Several LDs in WT cells were highly mobile, whereas KO cells contained large perinuclear LDs that were static (Fig 6C). Despite this, in KO cells, few mobile LDs could still be detected at the periphery of the cell. When we quantified the speed or distance travelled by these LDs, we could not detect any difference between WT and KO cells (Fig 6D). Thus, we conclude that lack of spastin impairs the redistribution of LDs from the perinuclear region to the cell periphery upon glucose starvation but does not affect the speed of LDs that move. Importantly, only reexpression of wild-type spastin in clones M1M87 and M1M87low, but not of MBD-deficient mutants, was required to restore LD dispersion in KO MEFs, emphasizing the role of spastin binding to the MTs in this phenotype (Fig 6E). Again, overexpression of wild-type spastin M1 failed to rescue the phenotype, indicating that excessive severing close to the LDs is detrimental (Fig 6E).

Finally, if spastin is implicated in the movement of LD from the perinuclear location, a prediction would be that a higher number of LDs remain in contact with the ER in spastin KO cells. To visualize ER-LD contact sites, we loaded WT and KO MEFs with OA for 24 h and then starved them in the absence of OA and glucose for 7 h, followed by analysis by transmission electron microscopy (TEM) and tomography (Figs 7A and S7). This shorter time point allowed us to still detect LDs in WT cells because most small dispersed LDs are extracted by the fixation conditions used for TEM. As expected, KO cells showed a higher number of LDs which were prominently clustered. The perimeter of LDs was unchanged compared with WT cells (Fig 7B). We quantified both the number of LDs that were in direct contact with ER tubules, and the length of the contact between the ER and the LDs. Lack of spastin caused an increase in the number of LDs that showed a contact with the ER tubules (Fig 7C), and the average length of the contact between the two organelles was increased (Fig 7D). Tomography further confirmed longer ER contacts with LDs in KO as compared with WT cells (arrowheads in Fig S7).

We conclude that spastin-M1 regulates the dispersion of LDs from perinuclear region to the cell periphery in an MT-dependent manner.

## Discussion

The MT-severing protein spastin has been implicated in diverse and highly regulated processes characterized by rearrangement of the cytoskeleton (Sherwood et al, 2004; Trotta et al, 2004; Zhang et al, 2007; Yu et al, 2008; Connell et al, 2009; Riano et al, 2009; Stone et al, 2012; Allison et al, 2013; Vietri et al, 2015). Most studies have

---

stained for PLIN2. Scale bar: 5 μm. **(G, H, I)** WT and KO MEFs were transfected with GFP-HPos or GFP-ACSL3 and serum-starved for 24 h (0 min) or further incubated for 15 min with oleic acid. **(G)** Representative single plane confocal images of MEFs expressing GFP-HPos or GFP-ACSL3 at 0 min Scale bar: 12 μm. (H) Quantification of HPos or (I) ACSL3 punctae at 0 min and after 15 min of oleic acid treatment. Graphs show mean ± SEM and individual data points of three independent biological replicates (in total 70 cells/genotype/condition). Unpaired t test, *P < 0.05, **P < 0.01. See also Fig S5.

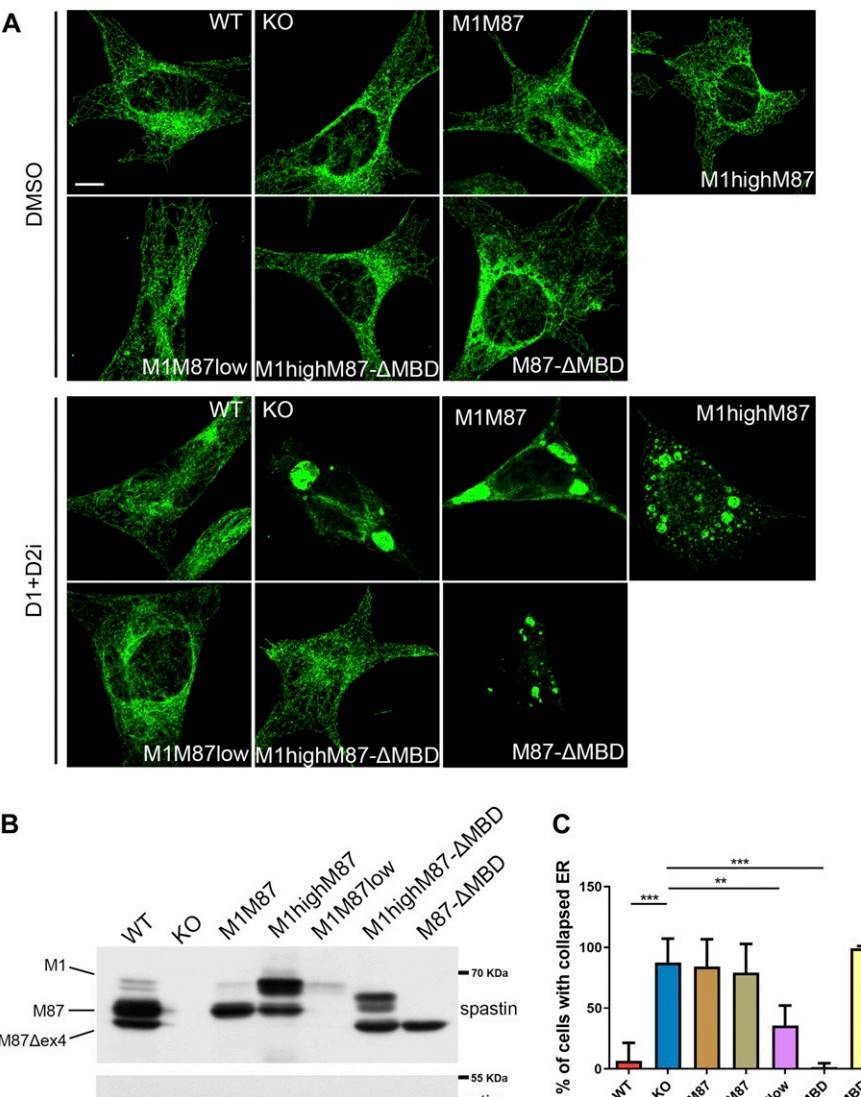

**Figure 5. Increased lipid droplet biogenesis buffers lack spastin-M1 at the ER.**
**(A)** MEF cell lines were starved in HBSS and treated either with inhibitors of DGAT1 and DGAT2 (D1+D2i) or DMSO overnight and stained with antibodies against reticulon 4. Representative single-plane confocal images. Scale bar: 10 $\mu$m. **(B)** Representative Western blot of spastin levels in the MEF cell lines used. **(C)** Bar graphs show the percentage of cells possessing collapsed ER in each cell line. Error bars represent SD. One-way ANOVA with post-Tukey test:**P < 0.01, ***P < 0.001. n = 4–5 independent experiments.

concentrated on the role of the most abundant spastin-M87 isoform, whereas the role of spastin-M1 has remained enigmatic, despite evidence for its enrichment in the nervous system and its relevance in the pathogenesis of HSP. Spastin-M1 localizes to the ER and the LDs and has been recently shown to tether LDs to peroxisomes (Papadopoulos et al, 2015; Chang et al, 2019). Here, we reveal additional MT-dependent and MT-independent roles of spastin-M1 in LD distribution and biogenesis.

We show that murine cells devoid of spastin accumulate LDs and TAGs independent from whether the source of fatty acids is exogenous or derives from autophagy, as it occurs upon starvation (Rambold et al, 2015). Our data point to the ER as one of the sources of LD accumulation. Lack of spastin leads to enhanced TAG synthesis and to an increased number of pre-LDs, defined as ER microdomains where initial steps of LD formation occur (Kassan et al, 2013). The curvature of the ER membrane and the local protein composition play permissive or restrictive roles for the formation of the lipid lens that represents the first step in LD formation (Walther et al, 2017; Olzmann & Carvalho, 2019). Spastin-M1 is embedded in the ER membrane via a transmembrane domain that is interrupted by a hydrophilic residue (R65 in human spastin), which is required for targeting the protein to LDs, allowing a hairpin configuration (Papadopoulos et al, 2015). Consistently, spastin-M1 specifically labels the regions of the ER where LDs emerge (Papadopoulos et al, 2015). A direct role of spastin-M1 in limiting pre-LD formation is suggested by a study that tested the ability of a peptide comprising spastin-M1 membrane-spanning sequence to promote phospholipid transbilayer movement (flip-flop). Whereas the wild-type peptide increased the flip rate of lipids in artificial membranes, an R65A mutation abolished this ability (Nakao et al, 2016). It is conceivable that cells lacking spastin may have an excess of phospholipids at the cytoplasmic leaflet of the ER membrane,

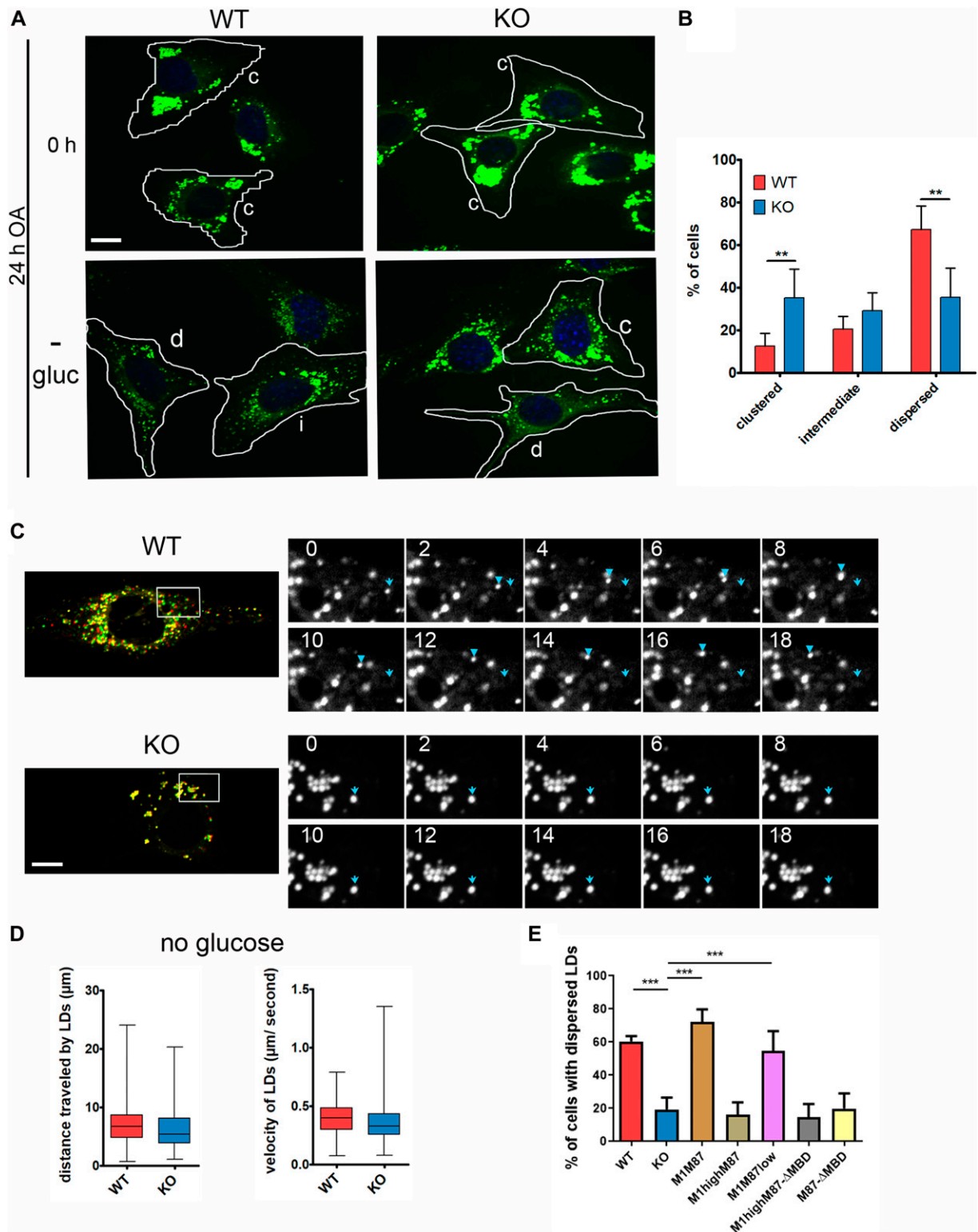

Figure 6. Spastin mediates dispersion of lipid droplets (LDs) from the ER.
(A) MEFs were fed for 24 h with oleic acid (0 h) followed by 16-h glucose starvation (–gluc) and stained with BODIPY 493/503. The cells were classified as dispersed (d), intermediate (i), or clustered (c), based on LD distribution. Dotted lines indicate cell border. Scale bar: 12 μm. (B) Distribution (%) of cells showing a dispersed, intermediate, or clustered phenotype (n = 5; ≈350 cells/genotype). Error bars represent SD. Unpaired t test, **P < 0.01, ***P < 0.001. (C) Merge of two frames 18 s apart of time-lapse movies taken after overnight glucose starvation. LDs in the first frame are labelled in red, whereas in last frame are labelled in green. LDs are highly mobile in WT, but stationary in KO cells. Scale bar: 12 μm (left). (D) Enlargements of boxed areas in (D), showing the individual frames (at 2-s intervals) of the movie. Blue arrow

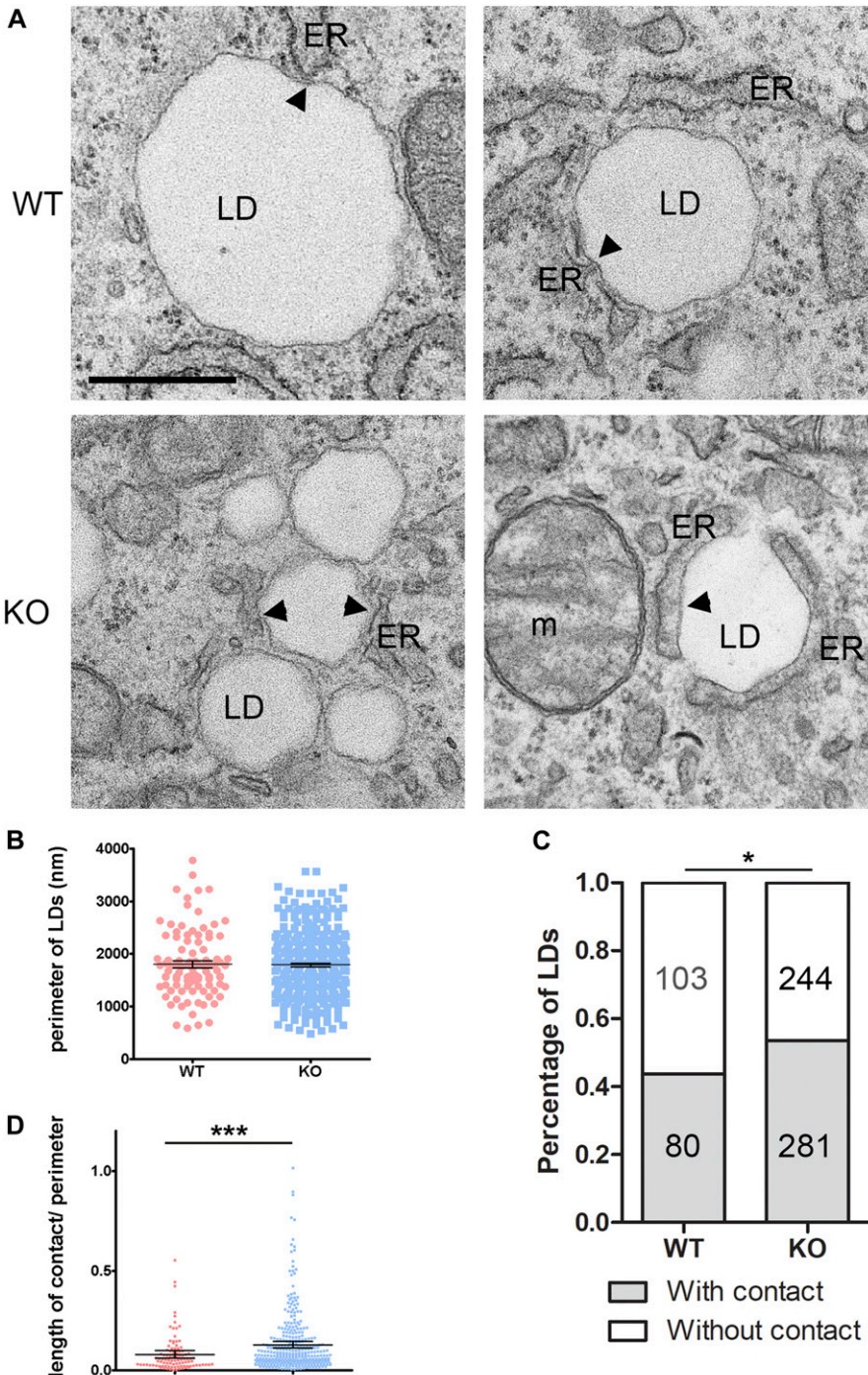

**Figure 7. *Spast* deletion leads to increased contact of ER with lipid droplets (LDs).**
**(A)** Representative TEM images of WT and KO MEFs treated with oleic acid for 24 h and starved in the absence of glucose for 7 h. Arrowheads show contact between ER and LDs. **(B)** Scatter plot of perimeter of LDs. Mean with 95% confidence interval are indicated. **(C)** Percentage of LDs with or without contact with the ER in WT and KO MEFs. Numbers inside bars represent the number of LDs quantified for each category (n = 20 cells for each genotype from two individual experiments). Fisher's exact test: *$P < 0.05$. **(D)** Scatter plot of length of contact of LDs with ER normalized to the perimeter of LDs (n = 20 cells of each genotype from two individual experiments). Mean with 95% confidence interval are indicated. Mann–Whitney test, ***$P < 0.001$. See also Fig S7.

which could favor LD emergence (Chorlay et al, 2019). Spastin-M1 may also shape the ER membrane in association with the fusogenic atlastin (Hu et al, 2009; Orso et al, 2009) and the curvature-promoting REEP1 (Voeltz et al, 2006) with which it interacts (Sanderson et al, 2006; Park et al, 2010). Interestingly, missense or loss-of-function mutations in these proteins have also been associated with LD abnormalities (Klemm et al, 2013; Falk et al, 2014; Renvoise et al, 2016; Zhao et al, 2016).

represents starting point of the traced LD, whereas the arrowhead marks its movement in subsequent frames (right). **(D)** Quantification of distance travelled and velocity of moving LDs in time-lapse movies taken after overnight glucose starvation in WT and KO MEFs. Data are shown as box and whisker plots (≈150 LDs from 30 cells/genotype: n = 3 independent experiments). **(E)** Bar graph shows the % of cells with dispersed LDs in the indicated MEF cell lines treated as in (A). Error bars are SD. One-way ANOVA with post-Tukey test: ***$P < 0.001$. n = 4 independent experiments. See also Fig S6.

In addition to a permissive role of spastin-deficiency to LD formation, a slightly increased autophagic flux in spastin KO cells under basal conditions likely contributes to deliver fatty acids for TAG synthesis. Intriguingly, in the absence of spastin, increased LD biogenesis may act as a buffering system to preserve the morphology of the ER, which is highly perturbed when synthesis of TAGs is prevented and cells are starved. Altered ER morphology has also been shown in yeast cells devoid of LDs upon starvation, a phenotype that could be recovered by inhibiting de novo fatty acid synthesis or inducing ER phospholipid synthesis (Velázquez et al, 2016). LDs are essential in mammalian cells to maintain the homeostasis of the ER and prevent ER stress and mitochondrial damage (Chitraju et al, 2017; Nguyen et al, 2017; Olzmann & Carvalho, 2019). Thus, increasing TAG synthesis and storing them in LDs may be a beneficial mechanism to overcome the loss of spastin-M1. Intriguingly, susceptibility of spastin KO cells to DGAT inhibitors was rescued only when a sufficient amount of spastin-M1 was reexpressed, independent of its ability to bind MTs. These data hint to a shaping function of spastin-M1 distinct from its enzymatic activity and are in agreement with the haploinsufficiency of spastin in human pathology. Further experiments will be necessary to fully understand the cause of the ER collapse phenotype and the mechanistic role of spastin and other ER morphogens that interact with spastin at the ER. In addition, the impact of autophagy on the ER phenotype remains to be investigated.

A recent study has suggested another MT-independent function of spastin-M1, which behaves as a tether between LDs and peroxisomes, allowing the supply of very long fatty acid for β-oxidation (Chang et al, 2019). Because this study has been performed upon overexpression of spastin-M1, it would be interesting to assess LD number and size under this condition. However, the role of spastin as a tether between LDs and peroxisomes is not in contradiction with our data. Lack of contacts between ER-associated LDs and other organelles may affect ER morphogenesis, and defects in channeling TAGs to peroxisomes can contribute to LD accumulation. Notably, the tethering domain of spastin overlaps with the MBD (Chang et al, 2019), suggesting a possible competition between binding peroxisomes or the MTs.

Besides affecting LD biogenesis at the ER, spastin plays a role in mediating LD dispersion from the perinuclear region to the cell periphery upon glucose deprivation, a condition that stimulates LD transport along MTs (Herms et al, 2015). Energy deprivation activates AMP activated protein kinase (AMPK) which results in a transient increase in detyrosinated MTs (Herms et al, 2015), which are a preferred substrate for spastin MT-severing activity (Roll-Mecak & Vale, 2008). Consistently, spastin competence to bind MTs is necessary to mediate LD dispersion. Moreover, reexpression of less-than-endogenous levels of wild-type spastin was enough to rescue the perinuclear accumulation of LDs, in agreement with the enzymatic role of the protein. Our data do not allow to uniquely attribute to spastin-M1 function because even low amounts of spastin-M87, recruited to the ER by oligomerizing with spastin-M1, may be sufficient to restore LD movement. Elegant work has previously shown that tubulin polyglutamylation controls MT severing by acting as a rheostat and activating or inhibiting spastin activity depending on the number of glutamate molecules (Lacroix et al, 2010; Valenstein & Roll-Mecak, 2016). We speculate that metabolic rewiring upon starvation leads to a change in the tubulin posttranslational

code that activates spastin MT-severing activity in the vicinity of LDs, facilitating their detachment from the ER required for long-distance travel along MTs. Consistent with this hypothesis, we found that LDs in cells lacking spastin display increased contact sites with the ER. However, this result could also reflect the perinuclear accumulation of LDs in KO cells and, hence, the higher density of ER sheets in this region. A role of spastin-mediated MT severing in LD trafficking may explain our previous findings of decreased LD number and TAG levels in Drosophila motor axons, upon down-nregulation of Dspastin (Papadopoulos et al, 2015). Importantly, these organisms lack a spastin isoform targeted to the ER, and therefore, phenotypes related to LD distribution and mobilization along MTs may be prevalent.

Our data further emphasize that tightly regulated levels of spastin-M1 are essential for ER and LD homeostasis because clones that overexpress MT-severing–competent spastin-M1 above endogenous levels displayed susceptibility to DGAT inhibition and failed to disperse LDs. Indeed, synthesis of spastin-M1 is restricted by a poor Kozak consensus around the first start codon (Claudiani et al, 2005). Although our data support both MT-dependent and MT-independent roles of spastin, a limitation of the rescue experiments is that they rely on single clones. Therefore, it will be important to further substantiate these studies with independent strategies.

A challenging question is whether the LD abnormalities that we observe in the immortalized *Spast* KO NSC34 cells also occur in cortical motor axons of human patients carrying mutations in the *SPAST* gene. So far, very little is known on the function and behavior of neuronal LDs. Neurons and axons have apparently a limited ability to store TAGs in LDs; however, very small LDs may be difficult to detect with current methods and play unrecognized roles in these cells (Pennetta & Welte, 2018). TAGs in neurons are likely to undergo a constant turnover, as loss of *Ddhd2*, which encodes a TAG hydrolase, leads to the neuronal accumulation of LDs in the mouse (Inloes et al, 2014, 2018). Notably, mutations in *DDHD2* are associated with HSP (Schuurs-Hoeijmakers et al, 2012). Long corticospinal axons depend on continuous supply of phospholipids to maintain their membrane. In the case of mutations in *DDHD2*, it is conceivable that impaired TAG hydrolysis would limit the amount of diacylglycerols available for phospholipid synthesis. Consistently, TAG hydrolysis has been shown to be essential to support regeneration properties of peripheral axons (Yang et al, 2020). In the case of mutations in *SPAST*, or in other HSP genes that cause disrupted ER morphogenesis, rewiring of lipid metabolism to increase TAG synthesis and LD formation may rescue ER homeostasis but trigger defects in phospholipid synthesis that in the long term would affect membrane maintenance and vesicular trafficking, leading to degeneration of extremely long axons. Although this remains a speculation, testing this hypothesis will be important to unravel the contribution of disturbed lipid metabolism in the pathogenesis of HSP.

# Materials and Methods

### Cell lines and culture conditions

All components necessary for cell culture were bought from Invitrogen unless stated otherwise. NSC34 cells (Cashman et al, 1992)

and MEFs were cultured and maintained in DMEM containing 4.5 g/l glucose supplemented with 2 mM glutamine, penicillin (200 U/mL), and streptomycin (200 μg/ml), and 5% defined fetal bovine serum (GE Healthcare Hyclone) for NSC34 cells or 10% FetalClone III (GE Healthcare Hyclone). Pheonix-ECO HEK293 cells were cultured in DMEM supplemented with GlutaMAX, 10% fetal FCS, penicillin (200 U/ml), and streptomycin (200 μg/ml). For glucose and serum starvation, we omitted glucose or serum, respectively, and further supplemented the medium with pyruvate (1 mM) and nonessential amino acids. HBSS starvation was carried out with HBSS (with calcium and magnesium).

### CRISPR-Cas9 *Spast* KO in NSC34 cells

NSC34 cells lacking spastin were generated using CRISPR-Cas9 double-nicking strategy (Ran et al, 2013). The following gRNAs targeting exon 5 of the *Spast* gene were employed:

5′-*CACCG*TCTGGAGCAAGACCGGGACC-3′
5′-*CACCG*CACTGCAACTAGGCGCCCTG-3′.

gRNAs were cloned into pX335 (Addgene) using the overhangs necessary for incorporation into the *Bbs*I restriction site (shown in italics). Cells were transfected with the two gRNAs simultaneously and allowed to express the plasmids. After trypsinization, single cells were serially diluted onto each well of a 96-well plate and allowed to grow. Colonies deriving from a single cell were further chosen and analyzed for KO of *Spast*.

The edited genomic region of the *Spast* gene was amplified with the following oligos:

5′-TAAGGATCCAAAGTGGAGCAGTTCCGAAG-3′
5′-TAAAAGCTTTGTGGTAGCTGCAGGACCAG-3′

The PCR fragment was subcloned into Tia1l vector and several clones were sequenced.

### Quantitative RT-PCR

Total RNA was extracted from cells using TRIzol reagent and retrotranscribed with the SuperScript First-Strand Synthesis System (Thermo Fisher Scientific) according to the manufacturer's protocol. Quantitative RT-PCR was performed with SYBR Green Master Mix (Thermo Fisher Scientific) using QuantStudio 12K Flex Real-Time PCR System thermocycler (Applied Biosystems). Each reaction was performed as a triplicate, and at least three biological replicates were performed per experiment. Mouse *Spast* levels were measured using primers 5′-TCCAGAGTATCCATCGGTCATTC-3′ and 5′-AAGGAAGTCCACTGACCCAAAA-3′. *Gapdh* was used for normalization and amplified with the following primers: 5′-AGGTCGGTGTGAACG-GATTTG-3′ and 5′-TGTAGACCATGTAGTTGAGGTCA-3′. The fold enrichment was calculated using the formula $2^{(-\Delta\Delta Ct)}$.

### Spastin-deficient MEFs

Primary fibroblasts were prepared from embryonic day 14.5 *Spast* KO embryos (Kasher et al, 2009) and were immortalized using a plasmid encoding the large T antigen, SV40. This procedure was performed in accordance with European Union (EU directive 86/609/EEC), national (Tierschutzgesetz), and institutional guidelines and were approved by local authorities (Landesamt für Natur, Umwelt, und Verbraucherschutz Nordrhein-Westfalen, Germany).

### Cell transfections

1 μg of GFP-HPos or GFP-ACSL3 (gift from Albert Pol) or 1 μg of pDEST-mCherry-EGFP-LC3B (Pankiv et al, 2007) were transfected using GeneJuice (Millipore), according to the manufacturer's protocol. siRNAs against *Atgl* (mouse *Pnpla2*) were obtained from Thermo Fischer Scientific (si*Atgl*2: 5′-GAAGAUAUCCGGUGGAUGAtt-3′) and Dharmacon (si*Atgl*1: 5′-GGGAAGAAUGCCAGCGUCA-3′; 5′-GCACAUUUAUCCCGGUGUA-3′; 5′-CUACAGAGAUGGACUUCGA-3′; 5′-UGAAAGAGCAGACGGGUAG-3′), and down-regulation was carried out using Lipofectamine2000, according to the manufacturer's protocol, for 72 h.

### Retroviral transduction of KO cells to reintroduce spastin

For retroviral transduction, spastin cDNA was amplified from previously generated templates (Papadopoulos et al, 2015). The E442Q mutation was obtained by site-directed mutagenesis. The following forward primers were used (ATG in bold):

eKozak and eKozak-ΔMBD:
5′-GGGGACAAGTTTGTACAAAAAAGCAGGCTCTGTGA**ATG**AATTCTCCGGGT-3′
gKozak and gKozak-E442Q:
5′-GGGGACAAGTTTGTACAAAAAAGCAGGCTGCCACC**ATG**AATTCTCCGGGT-3′
M87, M87-E442Q, and M87-ΔMBD:
5′-GGGGACAAGTTTGTACAAAAAAGCAGGCTGCCCTC**ATG**GCAGCCAAGAGG-3′
The following reverse primer was used:
5′-GGGGACCACTTTGTACAAGAAAGCTGGGTCTTAAACAGTGGTATCTCCAAA-3′.

The different *Spast* cDNA fragments were cloned using Gateway Cloning (Invitrogen) into pDONR-221 vector by BP reaction. The BP reaction was then cloned into a pBABE-puro plasmid using the LR reaction. For retroviral transduction, Pheonix-ECO HEK293 cells were transfected with pBABE-puro plasmid (containing the appropriate cDNA) for 24 h after which the cell medium was replaced with the medium for NSC34 cells. For infection, NSC34 or MEF KO cells were incubated with the supernatant from the HEK293 cells in the presence of 4 μg/ml polybrene for 24 h. This infection was repeated twice. The cells were then selected with puromycin (3 μg/ml) for 3 d. NSC34 rescue cells were analyzed as a pool. MEFs were then serially diluted in a 96-well plate, and colonies deriving from a single cell were analyzed for reintroduction of spastin by Western blot, and the integrity of spastin was confirmed by sequencing. The clone M1M87 was obtained using the eKozak vector, whereas the clones M1highM87 and M1M87low using the gKozak vector. These clones were used for experiments at early passages (<10) because they tended to lose spastin expression over time.

## Cell treatments

OA (O1008-5G; Sigma-Aldrich) was complexed with BSA (A6003; Sigma-Aldrich) in ratio 6:1 and used at a concentration of 400 $\mu M$. For treatment with varying concentrations of OA, the cells were treated with either 1,200, 600, 400, or 300 $\mu M$ for 24 h. To block TAG biosynthesis, the cells were treated with 10 $\mu M$ each of DGAT1 (PF-04620110) and DGAT2 (PF06424439) inhibitors (PZ0207 and PZ0233, respectively; Sigma-Aldrich) overnight in HBSS. To block autophagy, the cells were treated with 5 mM 3-methyladenine (M92811; Sigma-Aldrich) or 100 nM Bafilomycin A (B1793; Sigma-Aldrich) or 30 $\mu M$ chloroquine (C6628; Sigma-Aldrich) for the amount of time mentioned in figure legends. For blocking fatty acid synthesis, the cells were treated with 10 $\mu M$ triacsin C (SIH-203; Biomol) for 6 h.

## Extraction of soluble proteins

Extraction of soluble proteins was carried out according to Cappelletti et al (2003). The cells were washed once with PEM buffer (85 mM PIPES buffer, pH 6.9, 10 mM EDTA, pH 8.0, 1 mM $MgCl_2$, 2M glycerol, and 1 mM PMSF). For extraction, the cells were incubated with PEM buffer with 0.1% Triton X-100 and freshly added protease inhibitor cocktail for 10 min at RT and washed away with PBS. The cells were collected and lysed for Western blot.

## Immunofluorescence

For indirect immunofluorescence, the cells were grown onto glass coverslips and fixed with 4% paraformaldehyde for 30 min. To maintain LD integrity, the cells were further incubated with 50 mM NH4Cl for 10 min and permeabilized with 0.5% saponin in PBS for 10 min. After incubated for 10 min in blocking solution (0.1% saponin and 10% serum in PBS), primary antibodies were diluted in antibody solution (0.1% saponin and 1% serum in PBS) and applied to cells for 2 h at RT or overnight at 4°C. The cells were washed three times with PBS, and secondary antibodies diluted in antibody solution were applied to the cells for 1 h at RT. Finally, the cells were washed once with PBS containing DAPI (0.5 $\mu g$/ml), twice with PBS alone, and then the samples were mounted using FluorSave Reagent (Calbiochem). When LDs were stained, BODIPY 493/503 (5 $\mu M$) was applied in the washing step together with DAPI for 30 min. BODIPY C16 (1 $\mu M$) was fed to the cells for 24 h, washed away, and replaced with medium as indicated in figure legends. For detection of spastin, the cells were permeabilized with 0.2% Triton X-100 for 10 min followed by blocking with 10% serum in PBS for 10 min. For detection of reticulon 4, the cells were fixed in 4% PFA at 37°C to maintain ER integrity. Antibodies were diluted in 1% serum in PBS and incubated as stated earlier.

## Light microscopy and quantification

Fluorescent images were acquired using a 63× NA 1.4 oil objective and Axio Imager M2 microscope equipped with Apotome 2 (Zeiss) and processed using AxioVision software (Figs 1 and S4) Alternatively, Spinning Disk confocal microscope (Ultraview Vox; PerkinElmer) using Plan-Apo Tirf 60× oil objective with NA 1.49 was used. Images were acquired using Volocity software (version 6.1;

PerkinElmer) (Figs 4, 6, and S4–S6). In addition, microscopy was carried out using Zeiss Meta 710 microscope with Plan-Apochromat 63× oil objective of NA 1.4. Images were acquired using ZEN 2009 software (Figs 2, 5, S1, and S3). For super resolution microscopy, a similar staining procedure was used as stated above and mounted with ProLong Gold Antifade and imaged with TCS SP8 gSTED 3×, Leica Microsystems using HC PL APO 93× glycerol objective with NA 1.3. Images were acquired using LAS X software (Fig 4). Images show individual Z-stacks or projected images, as indicated in figure legends. Brightness levels were equally adjusted for image presentation for all images in a panel using Photoshop CC. PLIN2-puntae and LD number, size, perimeter, and area were counted using the particle analysis feature of Fiji. Each image was individually manually thresholded to eliminate background staining, and the watershed binary tool was used to separate clustered LDs. Number of PLIN2-puntae or LDs was normalized to the number of cells by dividing the count by the number of nuclei. For quantification of autophagosomes and autolysosomes, the images were acquired with 0.25-$\mu m$ stack distance, and maximum projection of images were used for the quantification. Number of green and red particles were calculated with the particle analysis feature of Fiji. Images were manually thresholded. The number of green particles resulted in the autophagosome count. The number of autolysosomes were calculated by subtracting the number of green particles from the red.

## Time-lapse video microscopy and quantification

Live-cell imaging experiments were carried out at 37°C and 5% $CO_2$ in the respective medium and were carried out using spinning disk confocal microscope (Ultraview Vox; PerkinElmer). To track LD movement, MEF cells were grown on glass-bottom dishes (MatTek Corporation), treated for 24 h with OA, and further starved overnight in medium without glucose. BODIPY 493/503 was added 30 min before imaging, washed away, and imaged for 2 min, and images were captured every 2 s. LD distance travelled and velocity were measured using the Fiji plugin, MTrackJ.

## Antibodies

Dilutions of antibodies used are as follows: PLIN2 (Gp40; Progen) 1:1,000 for WB and 1:200 for IF, ATGL (2138S; Cell Signaling Technology) 1:1,000 for WB, GAPDH (MAB374; Millipore) 1:4,000 for WB, spastin (Errico et al, 2004) 1:500 or 1:1,000 for WB and 1:100 for IF, alpha tubulin (T9026; Sigma-Aldrich) 1:2,000 for WB, actin (MAB1501R; Merck) 1:2,000 for WB, detyrosinated tubulin (AB3201; Millipore) 1:1,000 for WB, LC3 (NB100-2220; Novus Biologicals) 1:2,000 for WB, p62 (H00008878-M01; Abnova) 1:2,000 for WB, and reticulon 4 (NB100-56681; Novus Biologicals) 1:200 for IF.

## Electron microscopy and tomography

ACLAR foil slips (Plano) were autoclaved, placed in a six-well dish, and MEF cells were seeded on the foils (400,000 cells per well). The cells were treated for 24 h with OA and then for 7 h in medium without glucose. The cells were fixed in glutaraldehyde fixation buffer (2% glutaraldehyde, 2.5% sucrose, 3 mM $CaCl_2$, and 100 mM Hepes, pH 7.4) for 1 h at RT, and post-fixed in osmium tetroxide

solution (1% osmium tetroxide, 1.25% sucrose, and 10 mg/ml po-tassium ferrocyanid in 0.1 M sodium cacodylate buffer) for 1 h on ice. Dehydration steps were performed in ice-cold 50%, 70%, 90%, and 100% ethanol for 7 min each on ice. Foils were embedded in EPON Resin (Sigma-Aldrich). 70-nm ultrathin sections were cut with an ultramicrotome (UC7 Ultramicrotome; Leica). Sections were placed on carbon-coated grids with mesh 100 (Science Service). The grids were counterstained in uranyl acetate in $H_2O$ (2 mg/ml) for 15 min at 37°C. Images were acquired using a JEOL transmission electron microscope (TEM-2100 Plus; JEOL) with OneView 16-megapixel camera and Gatan Digital Micrograph program, version 3.30.2017.0, at an acceleration voltage of 80 kV. 20 random cells per genotype per experiment (n = 2) were imaged. Number, perimeter, and contact of LDs with ER in TEM images were measured blindly to the genotype using Fiji (n = 2, 20 cells per genotype).

For tomography, ultrathin sections of 200 nm were cut using an ultramicrotome (UC7; Leica) and incubated with 10 nm protein A gold (CMC) diluted 1:20 in ddH$_2$O. Sections were stained with Reynolds lead citrate solution for 3 min. Tilt series were acquired from –65° to 65° with 1° increment on a JEM-2100 Plus Transmission Electron Microscope (JEOL) operating at 200 kV equipped with a OneView 4K 32 bit (Gatan) using SerialEM (Mastronarde, 2005). Reconstruction was done using IMOD (Kremer et al, 1996).

### Lipidomics

TAG species in NSC34 cells were quantified by nano-electrospray ionization tandem mass spectrometry (Nano-ESI-MS/MS). Cell pellets (~3 × 10$^6$ cells, n = 3) were homogenized in 300 $\mu$l of Milli-Q water using the Precellys 24 Homogenisator (Peqlab) at 6,500 rpm for 30 s. The protein content of the homogenate was routinely determined using bicinchoninic acid. Aliquots of the cell homog-enate equivalent to 700 $\mu$g of protein were diluted to 500 $\mu$l with Milli-Q water and mixed with 1.875 ml of chloroform/methanol/37% hydrochloric acid 5:10:0.15 (vol/vol/vol) and 20 $\mu$l of 4 $\mu$M d5-TG Internal Standard Mixture I (Avanti Polar Lipids). Conditions of lipid extraction and Nano-ESI-MS/MS analysis using the QTRAP 6500 mass spectrometer (SCIEX) have been previously described (Kumar et al, 2015). Detection of TAG species was conducted by scanning for the neutral losses of the ammonium adducts of distinct fatty acids: m/z 273 (16:0), m/z 299 (18:1), and m/z 301 (18:0). A mass range of m/z 750–1,100 kD was scanned using a collision energy of 40 eV. All scans were conducted in the positive ion mode at a scan rate 200 D/ s with a declustering potential of 100 V, an entrance potential of 7 V, and a cell exit potential of 14 V (Ozbalci et al, 2013). Mass spectra were processed by the LipidView Software Version 1.2 (SCIEX) for identification and quantification of lipids. Endogenous TAG species were quantified by normalizing their peak areas to those of the internal standards. The calculated TAG amounts were normalized to the protein content of the cell homogenate.

### TAG synthesis and lipolysis experiments

NSC34 cells were incubated with normal culture medium containing trace amounts of $^{14}$C-OA (375 nCi/ml, specific activity 53 mCi/mmol) supplemented with non-labeled OA/BSA to a final concentration of 600 $\mu$M for respective amount of time. For lipolysis experiment, the cells were treated like above for 24 h after which they were washed and treated with culture medium without FCS but containing 1% fatty acid–free BSA and 10 $\mu$M of triacsin C for 6 h. After respective incubations, the cells were washed with PBS containing 1% fatty acid–free BSA, collected, and lysed. The lysate was used for lipid extraction and protein content determination. In addition, medium of cells incubated with triacsin C was also collected for lipid extraction.

Lipid extraction was carried out according to Wessel and Flugge (1984). Briefly, methanol was added to cell lysates and spun down. This was followed by addition of chloroform and centrifugation. Finally, pure water was added and centrifuged. Lower phase was collected and dried under N$_2$ stream. Cellular (or medium) lipids were dissolved in methanol/chloroform solution and loaded on TLC plates. The TLC plates (silica 60 W F254; 20 × 20 cm; Merck) were pre-run in methanol, dried, and activated for 10 min at 110°C on a TLC plate heater (TLC plate heater III; CAMAG). Standards and samples were applied onto the plate starting on the same front followed by running in solvent 1 (acid methyl ester, 1-propanol, chloroform, methanol, and 0.25% KCl [ratio 3:3:3:2:1]) to 4 cm. The plate was then removed from the tank and dried. The plate was placed in a chamber containing solvent 2 (75% N-hexan, 23% diethyl ether, and 2% acetic acid) and allowed to migrate to 1 cm from the top of the plate. The plate was dried and placed with solvent 3 (N-hexan) and allowed to migrate to 1 cm from the top of the plate. The plate was then exposed to a film, and autoradiographs of TLC plates were detected using Typhoon Trio (Amersham Biosciences).

### Statistical analysis

To compare the means of two groups, a two-tailed paired or un-paired $t$ test was carried out as indicated in figure legends. $\chi^2$ test was performed to compare the distribution of two groups. One-way ANOVA with post-Tukey test was performed to compare the means of multiple groups. Fisher's exact test was used to compare the percentage of LDs with or without ER contacts in WT and KO cells. Mann–Whitney test was used to compare the length of LD-ER contacts between WT and KO cells. The number of independent experiments performed is specified in the corresponding figure legend.

## Supplementary Information

## Acknowledgements

The authors thank Esther Barth, Eva Cziudaj, and Fiona Edenhofer for technical help; Chrisovalantis Papadopoulos for generating *Spast* KO MEFs; Andrew Grierson for sharing the *Spast* KO mice; Albert Pol, Thomas Langer, and Natalia Kononenko for providing reagents; and the CECAD imaging facility for support. This work was supported by grants from the Tom Wahlig Foundation and the E-Rare Research Programme within the framework of the ERA-NET E-Rare 2 (Federal Ministry of Education and Research, Neu-rolipid, 01GM1408A) to EI Rugarli.

## Author Contributions

N Tadepalle: conceptualization, formal analysis, investigation, visualization, and writing—review and editing.

L Robers: conceptualization, investigation, visualization, methodology, and writing—review and editing.

M Veronese: investigation, visualization, and writing—review and editing.

P Zentis: data curation.

F Babatz: investigation and visualization.

S Brodesser: formal analysis, investigation, and writing—review and editing.

AV Gruszczyk: investigation.

A Schauss: conceptualization and supervision.

S Höning: conceptualization, supervision, and investigation.

EI Rugarli: conceptualization, supervision, funding acquisition, and writing—original draft, review, and editing.

## Conflict of Interest Statement

The authors declare that they have no conflict of interest.

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
