## [Reviewer comments · Life Science Alliance]

Life Science Alliance

Microtubule-dependent and independent roles of spastin in lipid droplet dispersion and biogenesis

Nimesha Tadepalle, Lennart Robers, Matteo Veronese, Peter Zentis, Felix Babatz, Susanne Brodesser, Anja Gruszczyk, Astrid Schauss, Stefan Honing, and Elena Rugarli

DOI: <https://doi.org/10.26508/lsa.202000715>

Corresponding author(s): Elena Rugarli, CECAD Research Center

Review Timeline:

Submission Date:	2020-03-26
Editorial Decision:	2020-03-27
Revision Received:	2020-04-06
Editorial Decision:	2020-04-07
Revision Received:	2020-04-09
Accepted:	2020-04-14

Scientific Editor: Andrea Leibfried

Transaction Report:

Please note that the manuscript was previously reviewed at another journal and the reports were taken into account in the decision-making process at Life Science Alliance.

Reviewer #1 Review

Comments to the Authors (Required):

This very interesting paper shows that spastin, which was previously shown to localize to LDs, plays a functional role in LD biogenesis. Spastin KO cells accumulate LDs, and display defects in LD biogenesis and LD distribution in the cell interior during nutrient shortage. In general the experiments are well conducted and conclusive. The study is comprehensive, well conducted, and potentially exciting. LDs are well-known to traffic on the microtubule cytoskeleton, but how this is regulated and relates to the spatial distribution of LDs during metabolic states remains poorly understood.

There are several concerns that need to be addressed. The most general concern is that while the first half of the paper is well organized and thorough, the latter half is less complete. Numerous comparative controls are omitted which are needed to fully understand the nature of the observations (see details below). Furthermore, the last few figures feel somewhat disconnected from one another.

The general conclusion is that Spastin is required for proper LD biogenesis and distribution within the cell, but the mechanisms that underlie this are not understood. Perhaps the most exciting observation is that MT-severing events occur in proximity of nascent LD budding. Consistent with this, Spastin KO cells show defects in these severing events, and nascent LD bud sites are more abundant in KO cells. While interesting, these observations remain correlative with one another at best. What mechanistic role(s) does spastin have in LD biogenesis, and why does perturbing MT severing cause changes in nascent LD bud sites? In the Discussion, the authors imply a crosstalk between ER-shaping proteins and Spastin in LD biogenesis. Does loss of Atlastin or REEP1 phenocopy some of the observations in Spastin KO cells? Providing some additional mechanistic connection between MT severing and LD biogenesis would greatly add to the impact of this potentially exciting story.

Here are some additional specific comments:

1. page 12: Triacsin C is an inhibitor of fatty acyl CoA synthetases, not acetyl-CoA synthetases
2. Figure 3 D: the 14-C OA experiment with Triacsin C is informative, and suggests that the rate of TAG lipolysis is similar in WT and KO cells. However, it is plotted as normalized TAG for each cell type. This is mis-leading, as there is more TAG in the KO cell line. It may be better to plot each as total TAG normalized to protein, then plot the 0-6hr delta as comparable as a separate histogram.
3. Figure 4D: This appears to have the same panel twice, but in the legend is supposed to have a box and whisker plot on the bottom.
4. Figure 6A/B: Spastin KO cells have more GFP-HPos positive foci, suggesting more nascent LD bud sites. This is true even before the addition of OA, implying that cells begin with simply more nascent LD bud sites. Kassan et al show that ACSL3 is a good marker for nascent LDs. Are there more ACSL3 foci in KO cells?
5. Figure 6C: the observation that EB3 can enrich at actively maturing/budding(?) LDs is exciting. Spastin KO clearly affects this, but what is the comparative effect with drugs? For example, how does this compare with taxol addition?
6. Figure 7A: The observation that ER and LD co-localization is increased in the Spastin KO is intriguing, but the resolution of light microscopy alone is not sufficient to conclude that these two organelles are more connected in the KO. Another higher resolution technique such as transmission EM is needed to truly make this conclusion. Even thin section TEM can show clear ER connections to LDs.
7. In Figure 7, the data indicate that Spastin KO cells have accumulated LDs in the peri-nuclear region, but the actual speed of LD mobility, when observed, is unaffected. The authors conclude that Spastin KO somehow impairs the movement of LDs to the cell periphery. The mechanism here is lacking. Why are LDs accumulated in the peri-nuclear region in the KO? The authors infer that this is due to the inability to dis-connect LDs from the ER network, but this is correlation at best. An alternative model is that nascent LD bud sites are simply more concentrated in the peri-nuclear region in Spastin KOs. What is the distribution of GFP-HPos foci like in relation to the nucleus in KO versus WT cells? One way to dissect this further may also be to monitor the LD sub-cellular distribution in Seipin KO or knock-down cells, which may also display defects in ER-LD contacts. Do Seipin KO cells have more clustered LDs after 16hr no glucose?

Reviewer #2 Review

Comments to the Authors (Required):

In this paper, Tadepalle et al report on increased lipid droplet (LD) biogenesis and accumulation of triglycerides in murine cell lines lacking the microtubule severing protein spastin. The role of microtubule dynamics in LD biogenesis is not well understood and the current manuscript provides some interesting observations in this area. However, several of the major conclusions appear not well grounded and supported by the current data. These relate in particular to the connection of nascent LDs to the ER and nascent LD motility and its relation to microtubule dynamics regulated

by spastin. There are also inconsistencies in data presentation, making experiments difficult to interpret. Detailed comments are provided below.

Major points:

1. The authors imply that spastin regulates the detachment of LDs from the ER. Based on the data, this is not a valid conclusion. In Figure 7, no evidence is shown that the LDs are more or less connected to the ER +/- spastin. Figs 7A and S6A are not convincing, because the resolution is not sufficient to make statements about LD-ER proximity. For this, 3D EM data would be needed. ER is also present in the periphery of cells, not only in the perinuclear region. In 7A, the data shown (Manders coefficient for ER overlapping with LD is increased in spastin KO cells) could simply reflect the increased LD abundance in the spastin KO cells in this condition. Higher quality data should be provided or the conclusion should be omitted.

2. Could the dispersion of LDs upon glucose deprivation be related to the TAG storage phenotype? It seems likely that spastin KO cells have increased TAG remaining at 16 h of glucose deprivation, as lipolysis was apparently not affected by spastin KO but TAG synthesis was increased. Can this explain the differential dispersion of LDs in WT vs KO cells? This could be tested using WT cells loaded with different amounts of TAGs and observing whether dispersion is delayed when more TAGs are present at start.

Regarding Fig. 7F: the data should be provided as normalized to WT 0 h only, to show how tyrosination changes in WT cells in response to stimuli.

3. The data on LD motility and EB3 comets appears confusing and Fig. 6C-D leaves several questions.

Nascent LDs are usually described as motile structures, and in the videos of this paper (2 sec frame rate) several LDs display high motility. EB3 comets are typically also frequent and short-lived events in cells. In Fig 6C-D LD imaging was done every 15 sec for 15 min of OA loading. This frame rate seems too slow. It is possible that some EB3 events and LD movements are missed.

The way the data is presented, it appears as if there is one EB3-event (EB3 comet)/LD and one LD dislocation event. Do the LDs move either not at all or only once during 15 min? How exactly was a "LD-dislocation event" defined? This is a new term introduced and should therefore be carefully described and analyzed.

4. Based on methods, it appears as that the EB3 comets are related only to newly forming LDs, but the example shown in Fig. 6C has a LD (HPOS puncta) from start to finish. Are there comets on existing LDs and what is the timing between LD formation and EB3 comet formation? It is also not clear what the timing in Fig. 6C is in relation to the oleic acid loading time.

If the EB3 events are only related to newly formed LDs, there is a potential problem in data interpretation, as a bigger fraction of the spastin KO LDs are not newly formed (Fig 6B) but rather pre-existing, which would then give the observed result (less EB3 comets next to LD in spastin KO) but would not indicate a direct involvement of spastin in the process.

In Fig. 6C-D, the n of LDs/cell is very small (about 4), implying rather stringent inclusion criteria. These are not described. Imaging with an ER marker should be included, as EB3 comet events might also influence ER tubules and this might influence LD movement.

Additional points:

5. The conclusion that nascent LDs accumulate (or sites of LD formation increase) at the ER in

spastin deficient cells is not supported by the data. In Fig. 6B, the total LD number increase in response to OA is similar in WT and KO, suggesting that the amount of newly made LDs generated in response to OA is similar. Based on the increased TAG synthesis in KO cells, they might generate more LDs upon OA stimulation but since the starting point (0 min) between WT and KO is different, this cannot be concluded from these data. To assess if spastin KO cells generate more nascent LDs upon OA stimulation, a more stringent lipid starvation protocol could be employed, to have a starting point without HPOS puncta in both WT and spastin KO cells.

6. The authors state that it is "unlikely that impaired lipolysis is the cause for LD accumulation". However, ATGL knockdown does seem to abrogate the difference in LD accumulation between WT and spastin KO cells (Fig 3B, siATGL bars not significant) implying that lipolysis might actually be involved.

7. Regarding presentation of TAG data in Fig. 4E, it would make more sense to show total TAG or the most abundant TAG (C18:1) rather than the minor C16:0 TAG.

8. The display items should be carefully scrutinized, e.g. panel in Fig. 4D seems to be duplicated.

Reviewer #3 Review

Comments to the Authors (Required):

The manuscript by Tadepalle and colleagues describes how manipulating the microtubule severing spastin affects lipid droplets, including their size, number, and intracellular distribution, in cultured cells. Mutations in spastin play critical roles in certain neurodegenerative diseases, and spastin's connection to LDs, though long known, remains mysterious. Thus, uncovering how spastin and LDs interact is of both fundamental and clinical interest. This manuscript presents a lot of intriguing observations, but unfortunately is largely descriptive and mechanistic insights remain limited. Many interesting lines of inquiry are started, but are not deeply developed or brought to a firm conclusion. The authors develop intriguing models, but do not provide sufficient evidence to flesh them out, and thus the proposed connections remain unproven. In addition, there are technical problems and some observations are overinterpreted or misinterpreted. Thus, in its current state, the manuscript is far too preliminary to make a compelling story. Even if the technical issues I raise below can be addressed, I am not convinced that this manuscript represents a study of broad enough interest for readers of this journal; once revised, it may be much better suited for a more specialized journal.

Throughout the manuscript, the authors employ different methods to assess lipid droplet accumulation. In some figures, they employ PLIN2 puncta, in some they measure the area occupied by lipid droplets, in others they measure TAG levels. That makes it really hard to compare between figures and treatment. The figures also jump back and forth between NSC34 cells and MEFs, and I could not discern why particular cells were chosen for a particular experiment. Given that the spastin KO phenotype seems somewhat different between those two cell lines (see below), it gets very confusing.

Similarly, the authors use different methods to induce lipid droplets, sometimes by starvation in HBSS medium, sometimes by oleate feeding, often without a specified rationale. This may confound their analysis because starvation-induced lipid droplet formation is due to autophagy of membranous organelles, so it is conceivable that microtubules and thus spastin modulate TAG synthesis by influencing the supply of fatty acids released via autophagy, for example, because

autophagosomes do not form as efficiently or don't fuse with lysosomes or somehow the transfer of fatty acids from phagolysosomes to lipid droplets requires a microtubule-dependent step. The experiments in Fig. S3 only partially address these possibilities.

On page 10, the authors conclude from ATGL knockdowns that LD accumulation in the spastin mutants is not the cause for LD accumulation (Fig. 3A, B). I believe their data support the opposite conclusion: without ATGL knockdown, wild-type and spastin KO accumulate different amounts of LDs (as measure by PLIN2 levels), with knockdown the two genotypes are similar and similar to spastin KO with normal ATGL. These data suggest that ATGL is indeed responsible for the difference between wild-type and spastin KO and that lipolysis is significant in the wild type but not the mutant.

The rescue experiments in Fig. 4 are hard to interpret because the expression levels vary dramatically between the various constructs (Fig. 4B). For example, gKozak-E442Q is much lower expressed than gKozak. Thus, it may not be surprising that in Fig. 4D the bars for gKozak-E442Q appear to be intermediate between the nice rescue with gKozak and the KO. From these data, it is not possible to convincingly conclude that gKozak-E442Q does not rescue. The authors need to somehow normalize expression to be able to make meaningful conclusions. Without such a quantitative analysis, I am not convinced that "the ability to bind and sever MT's plays a crucial role" in how spastin affects LDs (page 15).

I have a few additional minor issues with Fig. 4: In A, the sequence around the second AUG is not shown for the gKozak construct, implying that those sequences were abolished. However, I believe that is not true. In D and E, significance is only shown for some of the bars. The authors should at least comment whether absence of the star symbols means that there is no significant difference, e.g. are the values for gKozak-E442Q and gKozak significantly different from each other (even omitting the expression level issue for now)? Without this information, it is hard to know what many of the bars mean. In E, lower panel, why are gKozak and eKozak-deltaMBD compared to each other? These constructs differ in multiple ways from each other, making it hard to conclude that lack of the MBD domain is responsible for the difference observed. Also, in E, a particular species of TAG is shown; why was this particular species omitted from Fig. 1F? This variation makes the comparison between figures needlessly complicated. Finally, the fact that spastin-M87 shows a certain degree of phenotypic rescue (page 13) seems inconsistent with their eventual model that something happens specifically to lipid droplets and their release from the ER.

The analysis with the microtubule stabilization drugs is confusing (Fig. 5). For one, the text says that vinblastine selectively increased Ac-tubulin, but there is no indication of statistical significance in Fig. 5C, so it is hard to know if this increase is reliable. The analysis also remains at the surface as far as LD mechanisms. The two microtubule stabilizing drugs affect LD accumulation differently (Fig. 5E) and they have - consistent with the literature - differential effects on tubulin modifications. The analysis stops here. The authors don't address whether the distinct effects on tubulin modifications or on microtubule dynamics are responsible for the differential effects on LDs and how this relates to the spastin effect. It seems a new line of inquiry was started, but not pursued to the extent that one can derive important conclusions.

Why do HPos puncta persist in starved KO cells to such a high extent? Are these indeed all nascent LDs that are somehow arrested? If so, does that indicate a pre-existing change in the structure of the ER?

The movies on which Fig. 6 is based on are technically challenging and in principle well done.

However, I am not convinced that they help with understanding the global phenotypes in spastin KO cells. For one, only a minority of HPos puncta moves and for those, the EB3 comet association makes only a two-fold difference (from 26 to 51%). Is there any indication that such small effects can explain the global effects on redistribution or lipid content? It is also puzzling, that there are still plenty of EB3 comets in the mutants, so MT severing still happens, yet global redistribution (as examined in Fig. 7) is dramatically affected. Also, in KO cells, the LDs still move, their movements just happen to be associated less often with EB3 comets. What might be more interesting would be to determine whether overall movement of HPos puncta is down in KO cells compared to wild type, i.e. does lack of spastin cause less motion of them?

I also have trouble understanding how the authors imagine LD motion and severing to be related. Is it that severing allows growth of MT which pushes LDs associated with MTs away, or does severing relax constraints on the LDs and thus they can be moved by other means (e.g. elastic effects that are no longer constraints by the MT)? But why would the comets occur before the movement, rather than concomitant with it? The authors could connect the LD movements and severing directly by expressing wild-type and severing deficient forms of spastin in the KO cells and check if the association with comets only gets restored when severing competent spastin is expressed. However, such an experiment may currently be not feasible, given the variability in expression of spastin constructs and the relatively small effect on comets.

A final problem with the model is that in Fig. 6C, it is not obvious that there is a cut in the microtubule where the comet appears nor is the moving LD obviously attached to a microtubule. How does that go along with the model?

The idea that spastin promotes LD detachment from the ER is an intriguing one. But the data in Fig. 7B, C do not directly address this. It seems that it is the clustered LDs that fail to move, but that lack of movement might simply be because the LDs are clustered and not that they are stuck to the ER. Also, why would lack of detachment from the ER promote clustering of LDs? That is not obvious.

There is one piece of data that suggest greater ER-LD contacts in the spastin KOs, shown in Fig. 7a and Fig. S6A. However, why does only one of three different measures of overlap show significant differences (Fig. S6A)? Finally, the LDs are clearly much larger in the spastin KO cells. Intuitively, it makes sense that larger lipid droplets are more likely to have close contact to ER, since they take up more space and thus may even push ER tubules apart, especially if the ER is a relatively fine mesh. ER that's pushed out of the way by a big LD would presumably remain closely associated with the LD surface. These issues should be carefully addressed, possibly by modeling, before concluding that the difference in Mader's coefficient in these figures is meaningful.

Other issues:

Page 5, last line: "We obtained one clone that showed..." This seems to imply that only a single mutation was recovered. However, later in the same paragraph, six alleles are described. Was only one clone analyzed at the protein level? This is confusing. Also, analyzing some of the other alleles for the initial LD phenotypes would go a long way towards guarding against off-target effects. Finally, I don't understand the notation in Fig. S1E: presumably the numbers refer to nucleotides, but counted from where? What does "c." means? And which of these alleles was used for the analysis in the other experiments?

Fig 1F: Why are only some TAG species elevated and not others? This observation does suggest that it is not simply the rate of TAG synthesis that is affected (as implied in the eventual model, where higher TAG synthesis is explained by more efficient recruitment of DGAT enzymes due to increased ER contacts; I don't see why in that model some TAG species would be more affected than others), but rather the supply of specific fatty acids.

Page 8: The paragraph on this page is introduced by raising concerns about off-target gene editing. Looking at knock-outs obtained by different methods is in principle a good idea. However, the authors look at different cell types (MEFs versus immortalized motoneurons) and the phenotypes are different: lipid droplet are more numerous in mutant motoneurons (Fig. 1D), while in KO MEFs LD number is unaltered, but LD size is bigger (Fig. 2C,D). In both situations, LDs are somehow altered, but because the exact alterations are different, it is hard to conclude that phenotypes in motoneurons are not partially due to off-target effects. The rescue experiments later in the paper are a much better control.

Fig. 3A: It is unclear to me how these cells were treated, e.g. basal versus HBSS starvation. This makes it harder to compare which situation in other panels this best corresponds to.

Fig. 4C. The spastin signal is hard to see; it would be important to show the KO cells stained under the exact same conditions in parallel so that it is clearer what portion of this signal might be above background.

On pages 18 and 19, the definition of nascent LDs seems to change. GFP-HPos is initially introduced as a way to examine nascent LDs (page 18); on page 19, it is stated that HPos-positive puncta that grow during the time of imaging signify nascent LDs. I realize that this is not necessarily contradictory, but as written will be confusing to many readers.

Fig. 6E: Two significant digits are not warranted for the percentages shown since the total number of droplets examined is less than 100.

Fig. S1B: The legend should explicitly say that two different exposures of the same Western are shown. Also, the label 55-68 kDa is confusing; please put the two markers at appropriate positions along the y axis.

Point-by-point response to Reviewers' comments

Reviewer #1

There are several concerns that need to be addressed. The most general concern is that while the first half of the paper is well organized and thorough, the latter half is less complete. Numerous comparative controls are omitted which are needed to fully understand the nature of the observations (see details below). Furthermore, the last few figures feel somewhat disconnected from one another.

We have extensively rewritten the manuscript, added controls and new experiments, and improved the connections between figures and the logic presentation of data.

The general conclusion is that Spastin is required for proper LD biogenesis and distribution within the cell, but the mechanisms that underlie this are not understood. Perhaps the most exciting observation is that MT-severing events occur in proximity of nascent LD budding. Consistent with this, Spastin KO cells show defects in these severing events, and nascent LD bud sites are more abundant in KO cells.

While interesting, these observations remain correlative with one another at best. What mechanistic role(s) does spastin have in LD biogenesis, and why does perturbing MT severing cause changes in nascent LD bud sites?

To address these questions, we have performed additional experiments and developed single-cell MEF clones expressing spastin-M1 and M87 at different ratio. Using these clones, we show that the observed phenotype in spastin KO cells is the combination of different defects. We find that increased LD biogenesis is a mechanism to buffer lack of spastin. Inhibition of TAG synthesis using DGAT inhibitors leads to a collapse of the tubular ER in spastin KO cells upon starvation. Expression of spastin-M1- Δ MBD is sufficient to rescue this phenotype, pointing to the requirement of membrane-shaping effects of spastin-M1 independent from microtubule-binding (Figure 5). However, when we examine another LD phenotype, i.e. LD dispersion upon glucose deprivation, only spastin-M1 competent to bind microtubules can rescue LD clustering in KO cells (Figure 6). We propose a new model, according to which controlled levels of spastin-M1 are recruited to the ER at the nascent LD sites. Here, spastin-M1 would balance LD budding by the concerted opposite action of the hairpin domain that imparts a certain curvature to the membrane and by regulated microtubule severing. MT-severing facilitates LD dispersion. Once on LDs, spastin could participate in other functions such as peroxisome contacts as recently demonstrated (Chang et al. 2019 JCB). We have extensively rewritten the discussion to take into consideration the new data.

In the Discussion, the authors imply a crosstalk between ER-shaping proteins and Spastin in LD biogenesis. Does loss of Atlastin or REEP1 phenocopy some of the observations in Spastin KO cells?

Although these experiments may give interesting insights on how Atlastins or REEP1 affect LD biogenesis, we are unsure if comparing these phenotypes will really help us to understand the mechanistic function of spastin in this process. We therefore believe that addressing this point goes beyond the scope of the current study, and remains something to be discussed.

Providing some additional mechanistic connection between MT severing and LD biogenesis would greatly add to the impact of this potentially exciting story.

We believe that the additional experiments that we have described above help to clarify this point.

Here are some additional specific comments:

1. page 12: Triacsin C is an inhibitor of fatty acyl CoA synthetases, not acetyl-CoA synthetases

This has been corrected.

2. Figure 3 D: the 14-C OA experiment with Triacsin C is informative, and suggests that the rate of TAG lipolysis is similar in WT and KO cells. However, it is plotted as normalized TAG for each cell type. This is mis-leading, as there is more TAG in the KO cell line. It may be better to plot each as total TAG normalized to protein, then plot the 0-6hr delta as comparable as a separate histogram.

We have already done this, and we agree with the Reviewer that this is a better way to show the data. This analysis clearly shows consumption of TAGs both in WT and KO cells.

3. Figure 4D: This appears to have the same panel twice, but in the legend is supposed to have a box and whisker plot on the bottom.

Sorry, this was a mistake in the figure preparation. We have corrected this in the new figure (Figure 2 of the revised manuscript).

4. Figure 6A/B: Spastin KO cells have more GFP-HPos positive foci, suggesting more nascent LD bud sites. This is true even before the addition of OA, implying that cells begin with simply more nascent LD bud sites. Kassan et al show that ACSL3 is a good marker for nascent LDs. Are there more ACSL3 foci in KO cells?

We have performed this experiment and we see an increase of ACSL3 foci in KO cells both after 24 h of serum deprivation and after 15 min of OA addition. The results are consistent with what we have seen using GFP-HPos (Figure 4G-I).

5. Figure 6C: the observation that EB3 can enrich at actively maturing/budding(?) LDs is exciting. Spastin KO clearly affects this, but what is the comparative effect with drugs? For example, how does this compare with taxol addition?

We do not imply that the mechanism by which vinblastine acts is the same as in spastin KO cells. We think that a thorough investigation of how these drugs act in regard to LD biogenesis is a completely independent study (see also response to Referee # 3). Therefore, we doubt that investing time in pursuing this line of investigation will tell us something on spastin mechanistic function. In view of the new data, we have decided to eliminate the experiments with the drugs from the revised version of paper.

6. Figure 7A: The observation that ER and LD co-localization is increased in the Spastin KO

is intriguing, but the resolution of light microscopy alone is not sufficient to conclude that these two organelles are more connected in the KO. Another higher resolution technique such as transmission EM is needed to truly make this conclusion. Even thin section TEM can show clear ER connections to LDs.

We agree with the Reviewer and have now performed TEM under conditions of glucose deprivation, to induce LD dispersion, and we have measured the % of LDs in contact with the ER, and the extent of contact surface between ER and LD in WT and KO cells. This experiment has some caveat since the dispersed small LDs are extracted by the fixation and are lost from our analysis. This experiment has however confirmed that the number of LD in contact with the ER and the length of contact between the ER and LDs is higher in the KO cells compared to the WT (Figure 7 of revised manuscript).

7. In Figure 7, the data indicate that Spastin KO cells have accumulated LDs in the peri-nuclear region, but the actual speed of LD mobility, when observed, is unaffected. The authors conclude that Spastin KO somehow impairs the movement of LDs to the cell periphery. The mechanism here is lacking. Why are LDs accumulated in the peri-nuclear region in the KO? The authors infer that this is due to the inability to dis-connect LDs from the ER network, but this is correlation at best. An alternative model is that nascent LD bud sites are simply more concentrated in the peri-nuclear region in Spastin KOs. What is the distribution of GFP-HPos foci like in relation to the nucleus in KO versus WT cells?

We do not see a preferential perinuclear localization of GFP-foci in KO cells. However, it is possible that both deficient LD dispersion and increased LD biogenesis contribute to this phenotype.

One way to dissect this further may also be to monitor the LD sub-cellular distribution in Seipin KO or knock-down cells, which may also display defects in ER-LD contacts. Do Seipin KO cells have more clustered LDs after 16hr no glucose?

This is an interesting question per se, but we doubt that this will help in illustrating spastin function.

Reviewer #2:

Major points:

1. The authors imply that spastin regulates the detachment of LDs from the ER. Based on the data, this is not a valid conclusion. In Figure 7, no evidence is shown that the LDs are more or less connected to the ER +/- spastin. Figs 7A and S6A are not convincing, because the resolution is not sufficient to make statements about LD-ER proximity. For this, 3D EM data would be needed. ER is also present in the periphery of cells, not only in the perinuclear region. In 7A, the data shown (Manders coefficient for ER overlapping with LD is increased in spastin KO cells) could simply reflect the increased LD abundance in the spastin KO cells in this condition. Higher quality data should be provided or the conclusion should be omitted.

We thank the Reviewer for this comment. As already mentioned (Reviewer 1 point 6), we have performed TEM under conditions of glucose deprivation, to induce LD dispersion, and we have measured the % of LD in contact with the ER, and the extent of contact surface between ER and LD in WT and KO cells. This experiment has some caveat since the dispersed small LDs are extracted by the fixation and are lost from our analysis. This experiment has however confirmed that the number of LD in contact with the ER and the length of contact between the ER and LDs is higher in the KO cells compared to the WT (Figure 7 of the revised manuscript). We have also added a tomography picture showing some examples of contacts (Figure S7).

2. Could the dispersion of LDs upon glucose deprivation be related to the TAG storage phenotype? It seems likely that spastin KO cells have increased TAG remaining at 16 h of glucose deprivation, as lipolysis was apparently not affected by spastin KO but TAG synthesis was increased. Can this explain the differential dispersion of LDs in WT vs KO cells? This could be tested using WT cells loaded with different amounts of TAGs and observing whether dispersion is delayed when more TAGs are present at start.

As suggested by the Reviewer, we have loaded cells with different amounts of OA followed by glucose deprivation. We observe clustering of LDs in KO cells in all conditions (Figure SC, D). Therefore, we can exclude that LD clustering is simply caused by more LDs at the beginning of the starvation. Instead, our new data show that the dispersion and the increased TAG synthesis are two different phenotypes (see also reply to Reviewer 1).

Regarding Fig. 7F: the data should be provided as normalized to WT 0 h only, to show how tyrosination changes in WT cells in response to stimuli.

We agree and the data are represented in this way in Figure S6A-B.

3. The data on LD motility and EB3 comets appears confusing and Fig. 6C-D leaves several questions.

Nascent LDs are usually described as motile structures, and in the videos of this paper (2 sec frame rate) several LDs display high motility. EB3 comets are typically also frequent and short-lived events in cells. In Fig 6C-D LD imaging was done every 15 sec for 15 min of OA loading. This frame rate seems too slow. It is possible that some EB3 events and LD movements are missed. The way the data is presented, it appears as if there is one EB3-event (EB3 comet)/LD and one LD dislocation event. Do the LDs move either not at all or only once during 15 min? How exactly was a "LD-dislocation event" defined? This is a new term introduced and should therefore be carefully described and analyzed.

The video of the paper showed LD movement in conditions of glucose deprivation, which enhances LD motility. Instead, the imaging of nascent LDs with HPos was performed after starving the cells with serum and readdition of OA. Under these conditions the LDs move very little, as they are mainly still connected with the ER. In the original submission, we defined a LD as nascent only when HPos puncta grew in size from the beginning of the movie, indirectly indicating accumulation of TAGs (as in the example in the old Figure 6C). HPos puncta showing this behavior were not many in each cell. A LD dislocation event was determined as a clear change of position of the LD, but we agree with the Reviewer that this definition is too generic and needs to be specified. Furthermore, we could not distinguish if this was a movement of a LD still connected to the ER. We took movies at 15 sec intervals,

because we experienced extensive bleaching with the EB3 signal. We have carefully considered our results and we agree with the Reviewer that this time frame is too long to really conclude that MT severing and LD movement are coupled or occurring sequentially. It is also possible that the reduced number of MT-severing events that we observed just reflected the more stable MT cytoskeleton upon loss of spastin. We have spent a lot of time to improve the imaging experiments and devise ways for automatic quantification. Imaging experiments at shorter intervals was hampered by the fact that the number of LDs showing a clear displacement was extremely low. We were unable to draw conclusions from such a limited set of data. Given the technical challenge of this experiment and the difficulty to visualize coupling of MT-severing with LD movements, we have decided to remove these data from the manuscript.

4. Based on methods, it appears as that the EB3 comets are related only to newly forming LDs, but the example shown in Fig. 6C has a LD (HPOS puncta) from start to finish. Are there comets on existing LDs and what is the timing between LD formation and EB3 comet formation? It is also not clear what the timing in Fig. 6C is in relation to the oleic acid loading time. If the EB3 events are only related to newly formed LDs, there is a potential problem in data interpretation, as a bigger fraction of the spastin KO LDs are not newly formed (Fig 6B) but rather pre-existing, which would then give the observed result (less EB3 comets next to LD in spastin KO) but would not indicate a direct involvement of spastin in the process. In Fig. 6C-D, the n of LDs/cell is very small (about 4), implying rather stringent inclusion criteria. These are not described. Imaging with an ER marker should be included, as EB3 comet events might also influence ER tubules and this might influence LD movement.

We have now removed these movies from the paper (see above).

Additional points:

5. The conclusion that nascent LDs accumulate (or sites of LD formation increase) at the ER in spastin deficient cells is not supported by the data. In Fig. 6B, the total LD number increase in response to OA is similar in WT and KO, suggesting that the amount of newly made LDs generated in response to OA is similar. Based on the increased TAG synthesis in KO cells, they might generate more LDs upon OA stimulation but since the starting point (0 min) between WT and KO is different, this cannot be concluded from these data. To assess if spastin KO cells generate more nascent LDs upon OA stimulation, a more stringent lipid starvation protocol could be employed, to have a starting point without HPOS puncta in both WT and spastin KO cells.

It is difficult to completely deplete spastin KO cells of sites of LD formation. A more stringent starvation protocol leads to cell death. Moreover, pre-LDs were originally described by the Pol lab (Kassan et al. JCB, 203, pages 985-1001, 2013) as: "restricted ER microdomains with a stable core of neutral lipids". Pre-LDs are resistant to starvation. We have specified this in the manuscript.

6. The authors state that it is "unlikely that impaired lipolysis is the cause for LD accumulation". However, ATGL knockdown does seem to abrogate the difference in LD accumulation between WT and spastin KO cells (Fig 3B, siATGL bars not significant) implying that lipolysis might actually be involved.

We realized that the quantification did not really reflect the western blots results. We have increased the number of experiments and requantified the blots. This confirmed our original conclusion (Figure 3G).

7. Regarding presentation of TAG data in Fig. 4E, it would make more sense to show total TAG or the most abundant TAG (C18:1) rather than the minor C16:0 TAG.

We agree, and we have now changed the figures to show all the main species.

8. The display items should be carefully scrutinized, e.g. panel in Fig. 4D seems to be duplicated.

We are sorry for this mistake. We have corrected this.

Reviewer #3

Throughout the manuscript, the authors employ different methods to assess lipid droplet accumulation. In some figures, they employ PLIN2 puncta, in some they measure the area occupied by lipid droplets, in others they measure TAG levels. That makes it really hard to compare between figures and treatment. The figures also jump back and forth between NSC34 cells and MEFs, and I could not discern why particular cells were chosen for a particular experiment. Given that the spastin KO phenotype seems somewhat different between those two cell lines (see below), it gets very confusing.

We have extensively revised the manuscript, and explained in detail the rationale for using different cell lines. Initially our experiments in NSC34 cells were dictated by the fact that these cells express higher levels of spastin-M1. Moreover, being of motoneuronal origin, these cells are more relevant for HSP. We also used MEFs for two reasons: i) to confirm data in an independent cell line; ii) because MEFs are flatter than NSC34 and are better suited for imaging experiments. In addition, we have improved the coherency of the methods used to analyze LD content in the different figures.

Similarly, the authors use different methods to induce lipid droplets, sometimes by starvation in HBSS medium, sometimes by oleate feeding, often without a specified rationale. This may confound their analysis because starvation-induced lipid droplet formation is due to autophagy of membranous organelles, so it is conceivable that microtubules and thus spastin modulate TAG synthesis by influencing the supply of fatty acids released via autophagy, for example, because autophagosomes do not form as efficiently or don't fuse with lysosomes or somehow the transfer of fatty acids from phagolysosomes to lipid droplets requires a microtubule-dependent step. The experiments in Fig. S3 only partially address these possibilities.

This is an important point, which we also considered. To carefully address if autophagosome formation and autophagosome-lysosome fusion occurs normally in spastin KO cells, we have performed experiments using a tandem-LC3 construct. These experiments have excluded problems in fusion of autophagosome with lysosomes occurring in the KO. In fact, KO cells showed a slightly increased autophagic flux. We conclude that in

absence of spastin, cells accumulate LDs independently whether the fatty acid source is exogenous or via autophagy.

On page 10, the authors conclude from ATGL knockdowns that LD accumulation in the spastin mutants is not the cause for LD accumulation (Fig. 3A, B). I believe their data support the opposite conclusion: without ATGL knockdown, wild-type and spastin KO accumulate different amounts of LDs (as measure by PLIN2 levels), with knockdown the two genotypes are similar and similar to spastin KO with normal ATGL. These data suggest that ATGL is indeed responsible for the difference between wild-type and spastin KO and that lipolysis is significant in the wild type but not the mutant.

We realized that the quantification did not really reflect the west, and we also discovered a possible pitfall in how the blots were quantified. To address this problem, we have increased the number of experiments and requantified the blots (Figure 3G). This confirmed our original conclusion.

The rescue experiments in Fig. 4 are hard to interpret because the expression levels vary dramatically between the various constructs (Fig. 4B). For example, gKozak-E442Q is much lower expressed than gKozak. Thus, it may not be surprising that in Fig. 4D the bars for gKozak-E442Q appear to be intermediate between the nice rescue with gKozak and the KO. From these data, it is not possible to convincingly conclude that gKozak-E442Q does not rescue. The authors need to somehow normalize expression to be able to make meaningful conclusions. Without such a quantitative analysis, I am not convinced that "the ability to bind and sever MTs plays a crucial role" in how spastin affects LDs (page 15).

The rescue experiments are performed using a mixed cell population transduced with the different viral vectors. In this context, each cell can have different expression level of the spastin protein, thus the western blot data reflect the whole cell population, but not necessarily the individual cells. We noticed that expression of the EQ mutants was extremely toxic and we could never reach high expression levels. For this reason, in the new rescue experiments in MEFs, we used clones deleted of the microtubule-binding domain which do not show this toxic effect. We have been successful in selecting clones in KO MEFs where we are re-expressing different ratio of spastin-M1 or spastin-M87 or mutants that cannot bind the microtubules. We selected clones with expression levels as similar as possible to the wild-type. We used these clones to: 1) assess rescue of ER phenotypes observed in KO cells upon DGAT inhibition; 2) assess rescue in condition of LD dispersion.

Using these clones, we show that the observed phenotype in spastin KO cells is the combination of different defects. We find that increased LD biogenesis is a mechanism to buffer lack of spastin. Inhibition of TAG synthesis using DGAT inhibitors leads to a collapse of the tubular ER in spastin KO cells upon starvation. Expression of spastin-M1- Δ MBD is sufficient to rescue this phenotype, pointing to the requirement of membrane-shaping effects of spastin-M1 independent from microtubule-binding (Figure 5). However, when we examine another LD phenotype, i.e. LD dispersion upon glucose deprivation, only spastin-M1 competent to bind microtubules can rescue LD clustering in KO cells (Figure 6). We propose a new model, according to which controlled levels of spastin-M1 are recruited to the ER at the nascent LD sites. Here, spastin-M1 would balance LD budding by the concerted opposite action of the hairpin domain that imparts a certain curvature to the

membrane and by regulated microtubule severing. MT-severing facilitates LD dispersion. Once on LDs, spastin could participate in other functions such as peroxisome contacts as recently demonstrated (Chang et al. 2019 JCB). We have extensively rewritten the discussion to take into consideration the new data.

I have a few additional minor issues with Fig. 4: In A, the sequence around the second AUG is not shown for the gKozak construct, implying that those sequences were abolished. However, I believe that is not true.

No, these sequences were not abolished. We have specified this in the Figure legend.

In D and E, significance is only shown for some of the bars. The authors should at least comment whether absence of the star symbols means that there is no significant difference, e.g. are the values for gKozak-E442Q and gKozak significantly different from each other (even omitting the expression level issue for now)? Without this information, it is hard to know what many of the bars mean.

We have included this information in all figure legends.

In E, lower panel, why are gKozak and eKozak-deltaMBD compared to each other? These constructs differ in multiple ways from each other, making it hard to conclude that lack of the MBD domain is responsible for the difference observed.

This was simply due to the successful establishment of the viral constructs. We think that what really matters is the amount of M1 or M87 expressed by each single construct. We have now removed these data, since we explore later in the manuscript the effect of MT-binding in MEFs.

Also, in E, a particular species of TAG is shown; why was this particular species omitted from Fig. 1F? This variation makes the comparison between figures needlessly complicated.

We realize that this is confusing, and have revised the figures to increase consistency.

Finally, the fact that spastin-M87 shows a certain degree of phenotypic rescue (page 13) seems inconsistent with their eventual model that something happens specifically to lipid droplets and their release from the ER.

Spastin-M87 shows only a partial rescue of the phenotype, as it shows values between WT and KO (Figure 2D, E).

The analysis with the microtubule stabilization drugs is confusing (Fig. 5). For one, the text says that vinblastine selectively increased Ac-tubulin, but there is no indication of statistical significance in Fig. 5C, so it is hard to know if this increase is reliable. The analysis also remains at the surface as far as LD mechanisms. The two microtubule stabilizing drugs affect LD accumulation differently (Fig. 5E) and they have - consistent with the literature - differential effects on tubulin modifications. The analysis stops here. The authors don't address whether the distinct effects on tubulin modifications or on microtubule dynamics are responsible for the differential effects on LDs and how this relates to the spastin effect. It

seems a new line of inquiry was started, but not pursued to the extent that one can derive important conclusions.

We have performed experiments with drugs in order to obtain further proof that stabilization of the MT cytoskeleton can affect LD content. However, this result remains largely correlative, and we agree with The Reviewer that this is a new line of investigation. We do not imply that the mechanism by which vinblastine acts is the same as in spastin KO cells. A thorough investigation of how these drugs act in regard to LD biogenesis is a completely independent study (see also response to Reviewer 1). Therefore, we doubt that investing time in pursuing this line of investigation will tell us something on spastin mechanistic function. We think that it may be better to remove completely these data, if they trigger more questions than answers.

Why do HPos puncta persist in starved KO cells to such a high extent? Are these indeed all nascent LDs that are somehow arrested? If so, does that indicate a pre-existing change in the structure of the ER?

We have carefully analyzed ER morphology in absence of spastin. Surprisingly, we do not observe a major phenotype; however, the combination of HBSS starvation and inhibition of LD formation using DGAT inhibitors triggers a dramatic collapse of ER only in KO cells (Figure 5). Thus, we think that increased LD biogenesis is fundamental in absence of spastin to support ER morphogenesis. Surprisingly, this phenotype could be rescued by re-expressing spastin-M1 without the MBD (Figure 5).

The movies on which Fig. 6 is based on are technically challenging and in principle well done. However, I am not convinced that they help with understanding the global phenotypes in spastin KO cells. For one, only a minority of HPos puncta moves and for those, the EB3 comet association makes only a two-fold difference (from 26 to 51%).

Is there any indication that such small effects can explain the global effects on redistribution or lipid content? It is also puzzling, that there are still plenty of EB3 comets in the mutants, so MT severing still happens, yet global redistribution (as examined in Fig. 7) is dramatically affected. Also, in KO cells, the LDs still move, their movements just happen to be associated less often with EB3 comets. What might be more interesting would be to determine whether overall movement of HPos puncta is down in KO cells compared to wild type, i.e. does lack of spastin cause less motion of them?

I also have trouble understanding how the authors imagine LD motion and severing to be related. Is it that severing allows growth of MT which pushes LDs associated with MTs away, or does severing relax constraints on the LDs and thus they can be moved by other means (e.g. elastic effects that are no longer constraints by the MT)? But why would the comets occur before the movement, rather than concomitant with it? The authors could connect the LD movements and severing directly by expressing wild-type and severing deficient forms of spastin in the KO cells and check if the association with comets only gets restored when severing competent spastin is expressed. However, such an experiment may currently be not feasible, given the variability in expression of spastin constructs and the relatively small effect on comets.

The time between frames in our movies was of 15 sec, so when we see a LD that has changed position, we do not know how close in time to the EB3 comet. The EB3 and the

displacement should not be concomitant in our opinion, but following each other. MT severing occurs, the new plus end recruit EB3 comet and only afterwards the LD can move. Whether it is because of less constraint or the attachment of a motor cannot be assessed with this experiment. For these reasons and also following concerns raised by this Reviewer and Reviewer # 2, we have decided to remove these data from the manuscript (see also response to Reviewer # 2, point 3) .

A final problem with the model is that in Fig. 6C, it is not obvious that there is a cut in the microtubule where the comet appears nor is the moving LD obviously attached to a microtubule. How does that go along with the model?

The idea that spastin promotes LD detachment from the ER is an intriguing one. But the data in Fig. 7B, C do not directly address this. It seems that it is the clustered LDs that fail to move, but that lack of movement might simply be because the LDs are clustered and not that they are stuck to the ER. Also, why would lack of detachment from the ER promote clustering of LDs? That is not obvious.

Based on our new data, we think that the LD clustering phenotype that we observed is a combination of increased biogenesis and defective dispersion.

There is one piece of data that suggest greater ER-LD contacts in the spastin KO cells, shown in Fig. 7a and Fig. S6A. However, why does only one of three different measures of overlap show significant differences (Fig. S6A)? Finally, the LDs are clearly much larger in the spastin KO cells. Intuitively, it makes sense that larger lipid droplets are more likely to have close contact to ER, since they take up more space and thus may even push ER tubules apart, especially if the ER is a relatively fine mesh. ER that's pushed out of the way by a big LD would presumably remain closely associated with the LD surface. These issues should be carefully addressed, possibly by modeling, before concluding that the difference in Mader's coefficient in these figures is meaningful.

We have performed TEM under conditions of glucose deprivation, to induce LD dispersion (as in Fig. 7), and we have measured the % of LD in contact with the ER, and the extent of contact surface between ER and LD in WT and KO cells. This experiment has some caveat since the dispersed small LDs are extracted by the fixation and are lost from our analysis. This experiment has however confirmed that the length of contact between the ER and LDs is higher in the KO cells compared to the WT (Figure 7 of revised manuscript).

Other issues:

Page 5, last line: "We obtained one clone that showed..." This seems to imply that only a single mutation was recovered. However, later in the same paragraph, six alleles are described. Was only one clone analyzed at the protein level? This is confusing. Also, analyzing some of the other alleles for the initial LD phenotypes would go a long way towards guarding against off-target effects. Finally, I don't understand the notation in Fig. S1E: presumably the numbers refer to nucleotides, but counted from where? What does "c." means? And which of these alleles was used for the analysis in the other experiments?

NSC34 cells are polyploid and therefore we found more than two mutated alleles. This has been clarified in the text. The numbering in Fig. S1E refers to the nucleotide position on the cDNA (according to official nomenclature for mutations). We have specified this in the text.

Fig 1F: Why are only some TAG species elevated and not others? This observation does suggest that it is not simply the rate of TAG synthesis that is affected (as implied in the eventual model, where higher TAG synthesis is explained by more efficient recruitment of DGAT enzymes due to increased ER contacts; I don't see why in that model some TAG species would be more affected than others), but rather the supply of specific fatty acids.

We do not have an explanation for this. It could be that species that are less abundant are measured less reliably. However, most species are increased.

Page 8: The paragraph on this page is introduced by raising concerns about off-target gene editing. Looking at knock-outs obtained by different methods is in principle a good idea. However, the authors look at different cell types (MEFs versus immortalized motoneurons) and the phenotypes are different: lipid droplet are more numerous in mutant motoneurons (Fig. 1D), while in KO MEFs LD number is unaltered, but LD size is bigger (Fig. 2C,D). In both situations, LDs are somehow altered, but because the exact alterations are different, it is hard to conclude that phenotypes in motoneurons are not partially due to off-target effects. The rescue experiments later in the paper are a much better control.

We agree and have now changed the order of the experiments to present all data with the NSC34 at the beginning of the paper followed by experiments in MEFs.

Fig. 3A: It is unclear to me how these cells were treated, e.g. basal versus HBSS starvation. This makes it harder to compare which situation in other panels this best corresponds to.

We have specified it in the text and Figure legend. Cells were cultured in basal conditions.

Fig. 4C. The spastin signal is hard to see; it would be important to show the KO cells stained under the exact same conditions in parallel so that it is clearer what portion of this signal might be above background.

Stainings of KO cells in the same conditions are shown in Figure S3.

On pages 18 and 19, the definition of nascent LDs seems to change. GFP-HPos is initially introduced as a way to examine nascent LDs (page 18); on page 19, it is stated that HPos-positive puncta that grow during the time of imaging signify nascent LDs. I realize that this is not necessarily contradictory, but as written will be confusing to many readers.

bart

We have revised the text to avoid this confusion. We adopt the designation of pre-LDs introduced by Kassan et al. (JCB, 2013).

Fig. 6E: Two significant digits are not warranted for the percentages shown since the total number of droplets examined is less than 100.

The Figure has been now removed from the manuscript.

Fig. S1B: The legend should explicitly say that two different exposures of the same Western are shown. Also, the label 55-68 kDa is confusing; please put the two markers at appropriate positions along the y axis.

We have revised this Figure.

Reviewer #1 Review

Comments to the Authors (Required):

This significantly revised paper probes the role of spastin-M1 variant in LD biogenesis and ER-LD interactions. Spastin KO cells display increased LDs in the cytoplasm and elevated TAG levels. Adding back M1 or M87 variants partially rescued this KO phenotype. Elevated LD levels appear to be due to a constitutive increase in TAG synthesis and LD biogenesis. Surprisingly, Spastin-KO cells exhibit altered ER morphology which can be rescued with certain variant clones. LD intracellular mobility is also impaired in Spastin KO cells, and can be rescued by re-introduction of WT but not microtubule-binding deficient Spastin mutants. Collectively the authors conclude that Spastin plays multiple roles in both ER homeostasis and LD motility. They conclude that Spastin plays both MT-dependent and independent roles in cell function.

This is an interesting and potentially exciting paper with many observations. Strengths include robust biochemical and cell biological assays. The first half of the paper is also easy to follow and read. This version is improved from the previous draft, and contains a wealth of data, but several problems remain.

The latter half of the paper still appears somewhat preliminary. Specifically, it is still unclear how Spastin has both MT-dependent and independent functions in maintaining ER morphology and LD positioning. The discrepancies between rescue phenotypes in ER morphology and LD motility are confusing and not addressed fully. Some of the conclusions in this section are also over-stated (see below), and need to be toned down. Addressing these concerns is necessary prior to publication.

General concerns:

1. Rescue of Spastin KO phenotypes: In Figure 5, the rescue expts with different spastin clones is confusing. Re-introduction of M1M87, which expresses both M1 and M87 variants at near endogenous levels, fails to rescue the ER morphology defect in Spastin KO cells. This is a simple complementation assay, so it is concerning that essentially a WT re-introduction does not rescue the KO. However, a variant over-expressing mutant M1 spastin that cannot bind to MTs DOES rescue. It is unclear from the information provided why this is the case. Later in Fig 6, the M1M87 clone is able to rescue the LD re-positioning phenotype (Fig 6E). Why this clone is able to rescue some but not other phenotypes requires more explanation. Is it possible that the ER collapse phenotype is from a different problem with the cells?

2. LD dispersion (severing?) conclusion: In Figure 7, the amount of contact LDs have with the ER is quantified from TEM (Fig 7C). Spastin KO cells manifest clustered LDs near the peri-nuclear space (which is rich in ER sheets). In line with this, ER-LD contacts are more extensive in KO versus WT. It is then concluded that Spastin is needed for LD dispersion. As it is written, it is unclear what the operational definition is for dispersion here. Is this meant to mean LD spatial positioning near the nuclear, or LD separation from the ER bilayer? Please clarify. In any case, this section needs to be toned down. Given that the peri-nuclear ER is rich in ER sheets, it is not surprising that there is more extensive ER-LD contacts.

3. Role of peroxisome turnover: Recent work indicates that Spastin acts as a tether connecting LDs to peroxisomes for fatty acid turnover (Chang, 2019). Spastin KO cells exhibit elevated TAG levels. Is it possible that, in addition to elevated TAG biogenesis, this may also partially be due to lower FA turnover in peroxisomes? Are there fewer peroxisomes in Spastin KO cells?

Minor comments:

Page 6: for the TAG lipidomics analysis in Fig 1D, please explicitly state whether the cells were exposed to oleate before analysis

Reviewer #2 Review

Comments to the Authors (Required):

In the revised manuscript of Tadepalle et al, the authors have retracted key mechanistic conclusions of the original submission and removed the related data.

In response to reviewer's major concerns 3 and 4, the authors overall response was that they were unable to strengthen their conclusions related to the role of spastin in severing microtubules in the proximity of nascent LD budding/prior to the movement of nascent LDs, and therefore they removed these data - and at the same time, a key mechanistic explanation for their observations.

In response to major point 2, the authors state that they have observed the clustering of LDs in KO cells in all conditions, referring to Figure SC, D. The authors probably mean Figure S6C, D, although this shows the data for WT cells and the classification of LD dispersion into clustered, intermediate and dispersed is apparently by eye only, leaving some space for uncertainty.

While removing key conclusions of the original manuscript, the authors have included intriguing but quite preliminary new data related to the susceptibility of the overall ER morphology of spastin KO cells to DGAT inhibition. These data are not sufficiently developed and would again necessitate additional investigations, such as exploring the effect of DGAT inhibition on ER morphology in WT vs KO cells in conditions other than HBSS (complete medium, oleic acid loading etc), and using more comprehensive analyses than reticulon-4 immunostaining. The entirely new conclusion - that spastin-M1 maintains the morphogenesis of the ER when TAG synthesis is prevented - thus raises major additional questions beyond the current revision.

Reviewer #3 Review

Comments to the Authors (Required):

The manuscript by Tadepalle and colleagues was dramatically restructured for this revision. These changes have addressed/eliminated a number of my previous concerns. However, some of the original concerns remain, and - more importantly - the manuscript still feels descriptive and preliminary. There are some important observations, e.g. that effects on dispersion and on ER shape under DGAT inhibition can be uncoupled, but mechanistic insight is limited. The manuscript nicely shows that lack of Spastin (using Spastin deletions, confirmed with rescue experiments) increases LDs in two different cell lines and provides evidence that this is likely attributable to increased TAG synthesis and enhanced formation of preLDs. It also shows that Spastin null cells are more sensitive to inhibition of TAG synthesis (as detected by ER morphology), show less LD dispersion throughout the cell, and display increased contacts with the ER, all valuable additions to the literature. However, in no case does the manuscript identify the molecular mechanism behind the phenotypes, other than testing whether Spastin's microtubule-binding domain is important. Therefore, I believe that in the current form this study neither fits the high standards of this journal nor is of interest to a broad audience of cell biologists.

The first two sections the manuscript nicely show that in NSC34 cells deletion of Spastin increases LDs (as assessed by PLIN2 Westerns, immunostaining, and TAG measurements) and that this phenotype can be rescued by viral expression of Spastin isoforms. These experiments are largely well done and documented, but I do see the following issues that need to be addressed (with points 3, 4 and 5 minor concerns)

1) The title of Fig. 2 claims that Spastin-M87 does not rescue LD and TAG accumulation. This is not supported by the data. The M87 cells are significantly different from the KO cells in the number of PLIN2 particles and the accumulation of several TAG species. As expression of the M87 isoform in these cells appears to still be lower than in the wild type, this partial rescue may simply reflect insufficient M87 levels. As I had pointed out in my previous review, this "seems inconsistent with

their eventual model that something happens specifically to lipid droplets and their release from the ER." Their rebuttal that the rescue is partial does not at all address this criticism and is therefore insufficient; if the non-ER targeted form can rescue some of the phenotypes, especially at levels close to or lower than observed in the wild type, this does suggest that the critical activity is not due to what happens at the ER. This requires an explanation.

2) Fig. 2 includes an analysis of E442Q mutants. As pointed out in my previous comments, these results are hard to interpret because the mutants are expressed at reduced levels. The authors seem to agree with this in the rebuttal letter, but they still leave the following sentences in the manuscript:

"Expression of mutant forms of spastin-M1 and/or spastin-M87 carrying the E442Q substitution in the Walker B motif of the AAA domain (gKozak-E442Q and M87-E442Q, Figure 2A, B), which abolishes ATPase activity, did not reduce the number of PLIN2-positive punctae and the TAG accumulation in KO cells (Figure 2D, E). However, these mutant forms were toxic and could not be expressed at the same levels as wild-type Spastin. These results provide a causal relationship between lack of spastin-M1 and the perturbation of lipid metabolism."

It seems odd to present data that they agree aren't fully interpretable. At the very least, the authors should explicitly acknowledge that the expression level problems do not allow them to draw firm conclusions.

3) I could not find any explanation for the band M87 delta ex4, in the legends or the text. I presume this represents an alternatively spliced isoform of M87. This needs to be explained. It also potentially undermines their conclusion that M87 doesn't (or doesn't fully) rescue. The Western suggests that this isoform is not expressed from their rescue construct; it is possible that this isoform contributes to the phenotype.

4) Glucose starvation alone does not induce PLIN2 accumulation in the mutant relative to the wild type in Fig. 1, but it does so in Fig. 4. Is there an explanation for this discrepancy between cell types?

5) Fig. 1C suggests that Spastin accumulates at the interfaces between droplets. Is that true? It would be helpful for the authors to comment on this curious pattern.

In Figs. 3 and S4, the authors nicely show that Spastin KO cells show increased TAG synthesis but no obvious changes in lipolysis. However, there are a number of curious observations that are not followed up. First, Fig. 3A shows not only increased TAG levels in KO cells, but also seems to indicate higher OA levels. Is that a real difference (i.e., one that holds up when normalized to protein levels in the sample)? If so, it might indicate that the phenotypes are due to increased uptake of exogenous FAs, and that the increase in TAG/LD synthesis is secondary. Second, the authors make a number of interesting observations for Spastin KO cells: ATGL levels as well as autophagic flux are increased. However, they provide no rationale for these observations nor do they follow up on them. These observations do suggest a broader disturbance of cellular metabolism in response to lack of Spastin; therefore, some of the phenotypes they observe might be quite indirectly caused by the absence of Spastin. For example, could the disturbances of ER morphology in the presence of DGAT inhibitors be a result of improper lipid balance due to altered autophagy? The implications of these observations need to be at least discussed. Finally, I have trouble interpreting the following observation "Despite this, block of autophagy concomitant with HBSS incubation either at early or late steps, using 3-methyladenine or bafilomycin A respectively, did not rescue the levels of PLIN2 that remained higher in KO compared to WT cells (Figure S4F, H)." Stopping FA influx from autophagy would lead to LD turnover in both genotypes, but because the Spastin KO cells start out with more LDs than the wild type they would presumably have higher LD levels at all time points, whether or not the initial increase in LD numbers was due to the increased autophagic flux. This is another instance where the authors describe a phenotype, but don't put it into any context.

In Fig. 4, the authors show that the basic observations from NSC34 cells also apply to MEFs, a good demonstration that this is a more general phenomenon. I only have some minor questions. It appears that Spastin expression (especially M1) is going up under HBSS treatment in MEFs, but not NSC34 cells. Is that relevant for your discussion? The mix of TAGs changing in MEFs seems to be different than in NSC34 cells. Is that a meaningful difference? If not, might it be better to show levels of all TAGs combined rather than divide them into classes. What is the reason for using super-resolution in panel F, rather than regular epifluorescence microscopy? If the authors propose that without that they couldn't distinguish clusters from large droplets, it is worth pointing that out. My main problem with this figure is that it does not go deeper. There are indeed more pre-LDs in the KO cells, but what does it mean? A reasonable interpretation is that this increase in pre-LDs is responsible for the increased TAG synthesis or the increased number of LDs. This could in principle be tested, e.g. if it is possible to reduce pre-LDs in the KO cells and show that TAG synthesis and LD number goes down. At a minimum, the authors should test whether they can rescue these phenotypes with the mutant constructs in Fig. 5 and determine if the same constructs rescue/fail to rescue both pre-LDs and TAG synthesis, for example.

The analysis of lack of Spastin in Fig. 5 is intriguing and reveals a novel connection between Spastin, ER and LDs. However, mechanistic insight remains modest, other than showing that microtubule binding is not required to rescue ER collapse. The authors conclude that this has something to do with the ER shaping function of Spastin, but they have no positive evidence for that hypothesis (e.g., pre-LDs in the absence of TAG might lose or acquire certain general ER proteins, in a Spastin dependent manner, which then indirectly affects ER morphology). The claim in the figure title "Increased LD biogenesis buffers lack of Spastin-M1 at the ER" is also not well supported. Yes, in the absence of all LD biogenesis, lack of Spastin-M1 somehow results in severe ER problems. However, it is not clear that the increased LD biogenesis in Spastin mutants prevents ER collapse; for that, the authors would have to reduce LD biogenesis to levels in the wild type and test whether this reduction (rather than full abolishment) is sufficient to cause ER collapse. It seems a critical question to address is why lack of Spastin causes increased LD biogenesis, but no mechanism is proposed. The situation is further complicated by the fact that rescue is so dependent on the exact levels of M1 and M87. The clone in which protein levels appear to be closest to wild type (M1M87), confusingly, shows NO rescue, and rescue is best when versions of M1 and M87 lacking the microtubule binding domain are expressed about equally, far from the wild-type situation. This pattern does not easily lend itself to a mechanistic interpretation, and the authors do not offer any, beyond describing that absolute levels and ratios seem to be important. Finally, M1highM87 and M1M87low appear to be clones derived from the same transfection. Given the widely different expression levels and ER phenotypes, it strikes me as prudent to test multiple clones with the same expression levels to make sure that the observed phenotypes are indeed correlated with expression levels and not a unique property of a particular clone.

Fig. 6 analyzes the role of Spastin in LD dispersion, and both section and figure title are overinterpreted. The figure title claims that Spastin mediates dispersion from the ER, but no analysis of the ER is presented in this figure. The section title claims that Spastin regulates dispersion, but there is no evidence of regulation; the observation is that in the absence of Spastin LDs dispersion is severely curtailed. One of the major conclusions of the section is that "lack of Spastin impairs the redistribution of LDs from the perinuclear region to the cell periphery upon glucose starvation but does not affect the speed of LDs that move." That nicely describes the phenotypes observed but does not explain why in the KO cells most LDs fail to move. Also, no data are provided as to the status of the microtubule cytoskeleton, other than that biochemically the fraction of non-tyrosinated tubulin seems normal. It would be important to know whether microtubule tracks are normal or altered (especially since the authors point out that non-

tyrosinated microtubules are a particularly good substrate for Spastin). Those data might help rule out certain models. The authors show that the speed of LD motion in the KO cells is normal. Lack of dispersion could come about by droplets having a tendency to move towards instead of away from the clusters, at normal speeds. Have the authors examined more details of the motility? The authors also conclude that microtubule binding is necessary for LD dispersion. Here, their data is overinterpreted: they do not test whether wild-type M87 promotes dispersion, therefore lack of dispersion in the M87 mutant cannot be interpreted. And wild-type M1highM87 fails to promote dispersion, so the absence of dispersion with mutant M1highM87 is also not unexpected. Given the complexity of expression levels/or M1-M87 ratios on phenotypes, it is conceivable that just the right ratio or levels of mutant M1 to mutant M87 would rescue. Finally, a minor point: the authors classify cells into clustered, intermediate or dispersed. It would be helpful to show examples of the intermediate state.

Fig. 7 summarizes lots of EM data that nicely show increased number and larger physical extents of LD/ER contacts in the KO cells. However, it is hard to know whether these increased contacts are a cause or consequence of the failed dispersion. In addition, in this case, they do not demonstrate that the phenotypes can be rescued by expression of wild-type Spastin.

March 27, 2020

RE: Life Science Alliance Manuscript #LSA-2020-00715-T

Prof. Elena I Rugarli
CECAD Research Center
Institute for Genetics
Joseph-Stelzmann-Str. 26
Koeln 50931
Germany

Dear Dr. Rugarli,

Thank you for transferring your revised manuscript entitled "Microtubule-dependent and independent roles of spastin in lipid droplet dispersion and biogenesis". Your manuscript was reviewed twice at another journal before, and the editors transferred those reports to us with your permission.

The reviewers appreciated your findings, but would have expected further reaching mechanistic insight. Lack thereof does not preclude publication here, and we would thus like to invite you to submit a final version of your manuscript to us. As we already discussed via phone, the remaining concerns should get addressed in a point-by-point response and by changing the manuscript text / data representation, leaving room for alternative explanations where needed and extending the discussion. As already discussed, it would be good to add data on peroxisomes in spastin KO cells.

A. FINAL FILES:

-- Summary blurb (enter in submission system): A short text summarizing in a single sentence the study (max. 200 characters including spaces). This text is used in conjunction with the titles of papers, hence should be informative and complementary to the title. It should describe the context and significance of the findings for a general readership; it should be written in the present tense

and refer to the work in the third person. Author names should not be mentioned.

B. MANUSCRIPT ORGANIZATION AND FORMATTING:

Sincerely,

Point-by-point response to Reviewers

Reviewer #1

This significantly revised paper probes the role of spastin-M1 variant in LD biogenesis and ER-LD interactions. Spastin KO cells display increased LDs in the cytoplasm and elevated TAG levels. Adding back M1 or M87 variants partially rescued this KO phenotype. Elevated LD levels appear to be due to a constitutive increase in TAG synthesis and LD biogenesis. Surprisingly, Spastin-KO cells exhibit altered ER morphology which can be rescued with certain variant clones. LD intracellular mobility is also impaired in Spastin KO cells, and can be rescued by re-introduction of WT but not microtubule-binding deficient Spastin mutants. Collectively the authors conclude that Spastin plays multiple roles in both ER homeostasis and LD motility. They conclude that Spastin plays both MT-dependent and independent roles in cell function.

This is an interesting and potentially exciting paper with many observations. Strengths include robust biochemical and cell biological assays. The first half of the paper is also easy to follow and read. This version is improved from the previous draft, and contains a wealth of data, but several problems remain. The latter half of the paper still appears somewhat preliminary. Specifically, it is still unclear how Spastin has both MT-dependent and independent functions in maintaining ER morphology and LD positioning. The discrepancies between rescue phenotypes in ER morphology and LD motility are confusing and not addressed fully. Some of the conclusions in this section are also over-stated (see below), and need to be toned down. Addressing these concerns is necessary prior to publication.

General concerns:

1. Rescue of Spastin KO phenotypes: In Figure 5, the rescue expts with different spastin clones is confusing. Re-introduction of M1M87, which expresses both M1 and M87 variants at near endogenous levels, fails to rescue the ER morphology defect in Spastin KO cells. This is a simple complementation assay, so it is concerning that essentially a WT re-introduction does not rescue the KO. However, a variant over-expressing mutant M1 spastin that cannot bind to MTs DOES rescue. It is unclear from the information provided why this is the case. Later in Fig 6, the M1M87 clone is able to rescue the LD re-positioning phenotype (Fig 6E). Why this clone is able to rescue some but not other phenotypes requires more explanation. Is it possible that the ER collapse phenotype is from a different problem with the cells?

We understand the Reviewer's concern and we were also initially surprised about this result. However, it should be noted that the clone M1M87 expresses the two spastin variants at endogenous ratio, but the absolute levels are still lower than in WT cells (especially for spastin-M1). In addition, in all our rescue experiments, spastin isoforms lacking exon 4 are still missing.

We have revised the Results to make clear these points:

Page 8:

"It should be noted that these constructs express only forms of spastin containing exon 4, and therefore do not fully reestablish endogenous spastin levels. So far, it is not known whether Δ ex4-spastin alternative isoforms play any specific functional role."

Page 12-13:

"We selected for further analysis clones expressing spastin-M1 and M87 at different levels and ratios: clone M1M87 expresses the two isoforms at nearly endogenous ratio, however the levels of both isoforms are lower than in wild-type conditions; clone M1highM87 expresses higher than endogenous levels of spastin M1; clone M1M87low synthesizes nearly endogenous levels of spastin-M1, but very low levels of spastin-M87 (Figure 5B)."

The fact that clone M1M87 is able to rescue the lack of motility suggests that when spastin works as a MT-severing enzyme its levels are not limiting (this is often the case for enzymes). However, in order to restore the ER collapse phenotype, endogenous levels of spastin-M1 appear to be necessary (in agreement with haploinsufficiency of *SPAST* in human patients). In this case, binding to MTs is not important, suggesting that this function of spastin-M1 is independent from its role as MT-severing enzyme. The situation is further complicated by the fact that excessive expression of MT-severing competent spastin-M1 is toxic, as already widely shown by data in the literature and by our rescue experiment with the clone M1highM87. Consistently, expression of spastin-M1 is restricted in vivo by several mechanisms, including a poor Kozak consensus sequence and an overlapping upstream open reading frame. We have revised the text both in the Results and Discussion sections to explain this better:

Results, page 13:

"Notably, expression in KO cells of high levels of spastin M1- Δ MBD, but not M87- Δ MBD was sufficient to rescue the susceptibility to DGAT inhibition (Figure 5A-C), indicating that restoring spastin-M1 at the ER independent of its MT-binding domain is sufficient to rescue the phenotype. Surprisingly, expression of MT-severing competent spastin-M1 did not rescue the ER collapse phenotype (Figure 5A-C), likely because spastin-M1 levels were insufficient (in clone M1M87) or too high (in M1highM87) (Figure 5A-C)."

Discussion, page 17 and 18:

"Intriguingly, susceptibility of spastin KO cells to DGAT inhibitors was rescued only when a sufficient amount of spastin-M1 was re-expressed, independent from its ability to bind MTs. These data hint to a shaping function of spastin-M1 distinct from its enzymatic activity and are in agreement with the haploinsufficiency of spastin in human pathology."..... "Consistently, spastin competence to bind MTs is necessary to mediate LD dispersion. Moreover, re-expression of less than endogenous levels of wild-type spastin was enough to rescue the perinuclear accumulation of LDs, in agreement with the enzymatic role of the protein".

Unfortunately, clone-specific effects cannot be completely ruled out. We have therefore added a note of caution in the revised discussion about limitations of the rescue experiments:

Page 18: *“While our data support both MT-dependent and independent roles of spastin, a limitation of the rescue experiments is that they rely on single clones. Therefore, it will be important to further substantiate these studies with independent strategies.”*

2. LD dispersion (severing?) conclusion: In Figure 7, the amount of contact LDs have with the ER is quantified from TEM (Fig 7C). Spastin KO cells manifest clustered LDs near the peri-nuclear space (which is rich in ER sheets). In line with this, ER-LD contacts are more extensive in KO versus WT. It is then concluded that Spastin is needed for LD dispersion. As it is written, it is unclear what the operational definition is for dispersion here. Is this meant to mean LD spatial positioning near the nuclear, or LD separation from the ER bilayer? Please clarify. In any case, this section needs to be toned down. Given that the peri-nuclear ER is rich in ER sheets, it is not surprising that there is more extensive ER-LD contacts.

We have carefully revised the Results text to describe the experiment as a dispersion of LDs from the perinuclear region to the periphery upon glucose deprivation. We have also revised the concluding sentence of this section as follows:

Page 15:

“We conclude that spastin-M1 regulates the dispersion of LDs from perinuclear region to the cell periphery in a MT-dependent manner.”

We analyzed ER contacts of all LDs in the examined cells by TEM, independently from their position. However, since KO cells have more perinuclear LDs, we cannot exclude a bias owing to the different density of ER sheets. We have therefore added the following sentence in the Discussion, page 18:

“Consistent with this hypothesis, we found that LDs in cells lacking spastin display increased contact sites with the ER. However, this result could in part reflect the perinuclear accumulation of LDs in KO cells, and hence the higher density of ER sheets in this region.”

3. Role of peroxisome turnover: Recent work indicates that Spastin acts as a tether connecting LDs to peroxisomes for fatty acid turnover (Chang, 2019). Spastin KO cells exhibit elevated TAG levels. Is it possible that, in addition to elevated TAG biogenesis, this may also partially be due to lower FA turnover in peroxisomes? Are there fewer peroxisomes in Spastin KO cells?

We have performed this experiment in MEFs and found that the number of peroxisomes is not significantly different between WT and KO cells. We add the figure below for the Reviewer. We have decided not to include this data in the manuscript because we feel that it interrupts the flow of the story and raises additional questions that are beyond the scope of the current study.

Figure Legend: The number of peroxisomes stained with an antibody against PMP70 is not significantly different between WT and KO in MEFs. Box plots showing the results in 68 cells coming from 3 independent experiments.

Minor comments:

Page 6: for the TAG lipidomics analysis in Fig 1D, please explicitly state whether the cells were exposed to oleate before analysis.

Cells were not exposed to oleate before analysis. We have specified it in the corresponding figure legend.

Reviewer #2

In the revised manuscript of Tadepalle et al, the authors have retracted key mechanistic conclusions of the original submission and removed the related data.

In response to reviewer's major concerns 3 and 4, the authors overall response was that they were unable to strengthen their conclusions related to the role of spastin in severing microtubules in the proximity of nascent LD budding/prior to the movement of nascent LDs, and therefore they removed these data - and at the same time, a key mechanistic explanation for their observations.

We have removed these data, following well-grounded criticisms from the Reviewer that we could not experimentally address in a satisfactory manner.

In response to major point 2, the authors state that they have observed the clustering of LDs in KO cells in all conditions, referring to Figure SC, D. The authors probably mean Figure S6C, D, although this shows the data for WT cells and the classification of LD dispersion into clustered, intermediate and dispersed is apparently by eye only, leaving some space for uncertainty.

We apologize for a mistake in the response letter. Indeed, we referred here to Fig. S6C, D. We have exactly done what the Reviewer has previously requested to rule out that simply loading WT cells with more OA would change the distribution of LDs and increase their clustering.

While removing key conclusions of the original manuscript, the authors have included intriguing but quite preliminary new data related to the susceptibility of the overall ER morphology of spastin KO cells to DGAT inhibition. These data are not sufficiently developed and would again necessitate additional investigations, such as exploring the effect of DGAT inhibition on ER morphology in WT vs KO cells in conditions other than HBSS (complete medium, oleic acid loading etc), and using more comprehensive analyses than reticulon-4 immunostaining. The entirely new conclusion - that spastin-M1 maintains the morphogenesis of the ER when TAG synthesis is prevented - thus raises major additional questions beyond the current revision.

We realize that these data are preliminary, however we believe that they are worth reporting. We have stated in the Discussion, page 17, that:

“Further experiments will be necessary to fully understand the cause of the ER collapse phenotype and the mechanistic role of spastin and other ER morphogens that interacts with spastin at the ER. In addition, the impact of autophagy on the ER phenotype remains to be investigated.”

Reviewer #3:

The manuscript by Tadepalle and colleagues was dramatically restructured for this revision. These changes have addressed/eliminated a number of my previous concerns. However, some of the original concerns remain, and - more importantly - the manuscript still feels descriptive and preliminary. There are some important observations, e.g. that effects on dispersion and on ER shape under DGAT inhibition can be uncoupled, but mechanistic insight is limited. The manuscript nicely shows that lack of Spastin (using Spastin deletions, confirmed with rescue experiments) increases LDs in two different cell lines and provides evidence that this is likely attributable to increased TAG synthesis and enhanced formation of preLDs. It also shows that Spastin null cells are more sensitive to inhibition of TAG synthesis (as detected by ER morphology), show less LD dispersion throughout the cell, and display increased contacts with the ER, all valuable additions to the literature. However, in no case does the manuscript identify the molecular mechanism behind the phenotypes, other than testing whether Spastin's microtubule-binding domain is important. Therefore, I believe that in the current form this study neither fits the high standards of this journal nor is of interest to a broad audience of cell biologists.

The first two sections the manuscript nicely show that in NSC34 cells deletion of Spastin increases LDs (as assessed by PLIN2 Westerns, immunostaining, and TAG measurements) and that this phenotype can be rescued by viral expression of Spastin isoforms. These experiments are largely well done and documented, but I do see the following issues that need to be addressed (with points 3, 4 and 5 minor concerns)

1) The title of Fig. 2 claims that Spastin-M87 does not rescue LD and TAG accumulation. This is not supported by the data. The M87 cells are significantly different from the KO cells in the number of PLIN2 particles and the accumulation of

several TAG species. As expression of the M87 isoform in these cells appears to still be lower than in the wild type, this partial rescue may simply reflect insufficient M87 levels. As I had pointed out in my previous review, this "seems inconsistent with their eventual model that something happens specifically to lipid droplets and their release from the ER." Their rebuttal that the rescue is partial does not at all address this criticism and is therefore insufficient; if the non-ER targeted form can rescue some of the phenotypes, especially at levels close to or lower than observed in the wild type, this does suggest that the critical activity is not due to what happens at the ER. This requires an explanation.

In these experiments, we observe that a clone expressing only spastin-M87 rescues the phenotype less efficiently than a clone expressing mainly spastin-M1. We agree with the Reviewer that this does not completely rule out a role of spastin-M87 and have therefore changed the titles of the corresponding Result section and of Fig. 2 in: "*Spastin re-expression rescues LD and TAG accumulation.*"

Furthermore, we have modified the last sentence of the corresponding Result section (page 8-9) as follows: "*These results provide a causal relationship between lack of spastin and the perturbation of lipid metabolism, and implicate spastin-M1 and to a lesser extent spastin-M87 in the phenotype.*"

In these experiments in NSC34 cells, the phenotype observed probably stems from a combination of defects, and is only in part caused by the function of spastin at the ER. This is supported by the experiments performed in MEFs.

2) Fig. 2 includes an analysis of E442Q mutants. As pointed out in my previous comments, these results are hard to interpret because the mutants are expressed at reduced levels. The authors seem to agree with this in the rebuttal letter, but they still leave the following sentences in the manuscript: "Expression of mutant forms of spastin-M1 and/or spastin-M87 carrying the E442Q substitution in the Walker B motif of the AAA domain (gKozak-E442Q and M87-E442Q, Figure 2A, B), which abolishes ATPase activity, did not reduce the number of PLIN2-positive punctae and the TAG accumulation in KO cells (Figure 2D, E). However, these mutant forms were toxic and could not be expressed at the same levels as wild-type Spastin. These results provide a causal relationship between lack of spastin-M1 and the perturbation of lipid metabolism." It seems odd to present data that they agree aren't fully interpretable. At the very least, the authors should explicitly acknowledge that the expression level problems do not allow them to draw firm conclusions.

We have modified the last sentence as follows: "*However, these mutant forms were toxic and could not be expressed at the same levels as wild-type spastin, preventing to conclusively assess the role of a functional AAA domain.*"

3) I could not find any explanation for the band M87 delta ex4, in the legends or the text. I presume this represents an alternatively spliced isoform of M87. This needs to be explained. It also potentially undermines their conclusion that M87 doesn't (or doesn't fully) rescue. The Western suggests that this isoform is not expressed from their rescue construct; it is possible that this isoform contributes to the phenotype.

The Reviewer is correct. These alternatively spliced isoforms of spastin are not expressed by our constructs. We now explicitly acknowledge this, when we describe the constructs: *“It should be noted that these constructs express only forms of spastin containing exon 4, and therefore do not fully reestablish endogenous spastin levels. So far, it is not known whether Δ ex4- spastin alternative isoforms play any specific functional role.”*

4)Glucose starvation alone does not induce PLIN2 accumulation in the mutant relative to the wild type in Fig. 1, but it does so in Fig. 4. Is there an explanation for this discrepancy between cell types?

As the Reviewer has noticed this is a different behavior in NSC34 and MEFs and we do not have an explanation for this discrepancy.

5)Fig. 1C suggests that Spastin accumulates at the interfaces between droplets. Is that true? It would be helpful for the authors to comment on this curious pattern.

To assess if this is the case, accurate quantification would be needed. We believe that the Reviewer is referring to Fig. 2C, which aims to show that spastin-M1 expressed by the retroviral vectors labels the LD surface, as expected. We think that it is beyond the scope of the current manuscript to further investigate this issue.

In Figs. 3 and S4, the authors nicely show that Spastin KO cells show increased TAG synthesis but no obvious changes in lipolysis. However, there are a number of curious observations that are not followed up. First, Fig. 3A shows not only increased TAG levels in KO cells, but also seems to indicate higher OA levels. Is that a real difference (i.e., one that holds up when normalized to protein levels in the sample)? If so, it might indicate that the phenotypes are due to increased uptake of exogenous FAs, and that the increase in TAG/LD synthesis is secondary.

No this is not a real difference. Below, we show the quantification of OA levels across experiments normalized to protein levels. There is no difference between WT and KO cells.

Figure Legend. Quantification of OA levels in experiments depicted in Figure 3A. Note that a different program has been used to quantify the OA band, so the absolute values cannot be compared with those of TAGs in Figure 3A.

Second, the authors make a number of interesting observations for Spastin KO cells: ATGL levels as well as autophagic flux are increased. However, they provide

no rationale for these observations nor do they follow up on them. These observations do suggest a broader disturbance of cellular metabolism in response to lack of Spastin; therefore, some of the phenotypes they observe might be quite indirectly caused by the absence of Spastin. For example, could the disturbances of ER morphology in the presence of DGAT inhibitors be a result of improper lipid balance due to altered autophagy? The implications of these observations need to be at least discussed.

We have added the following sentence to the Discussion:

“Further experiments will be necessary to fully understand the cause of the ER collapse phenotype and the mechanistic role of spastin and other ER morphogens that interacts with spastin at the ER. In addition, the impact of autophagy on the ER phenotype remains to be investigated.”

Finally, I have trouble interpreting the following observation "Despite this, block of autophagy concomitant with HBSS incubation either at early or late steps, using 3-methyladenine or bafilomycin A respectively, did not rescue the levels of PLIN2 that remained higher in KO compared to WT cells (Figure S4F, H)." Stopping FA influx from autophagy would lead to LD turnover in both genotypes, but because the Spastin KO cells start out with more LDs than the wild type they would presumably have higher LD levels at all time points, whether or not the initial increase in LD numbers was due to the increased autophagic flux. This is another instance where the authors describe a phenotype, but don't put it into any context.

We have slightly reformulated this paragraph to increase clarity:

“These data suggest an increase in the autophagic flux in absence of spastin, which is also supported by decreased levels of the autophagic adaptor p62 (Figure S4F, G). Thus, enhanced autophagy may contribute to deliver fatty acids for TAG synthesis in KO cells. Incubation of cells with HBSS and 3-methyladenine or bafilomycin A, which inhibit autophagy either at early or late steps, respectively, was however not sufficient to abrogate the increased levels of PLIN2 in KO compared to WT cells (Figure S4F, H)”.

In Fig. 4, the authors show that the basic observations from NSC34 cells also apply to MEFs, a good demonstration that this is a more general phenomenon. I only have some minor questions. It appears that Spastin expression (especially M1) is going up under HBSS treatment in MEFs, but not NSC34 cells. Is that relevant for your discussion?

We have sometimes observed this effect, however quantification across several experiments did not conclusively demonstrate that spastin expression responds to starvation.

The mix of TAGs changing in MEFs seems to be different than in NSC34 cells. Is that a meaningful difference? If not, might it be better to show levels of all TAGs combined rather than divide them into classes.

It is expected that the mix of detected TAGs would be cell-type specific, and so we are not entirely surprised that also the TAGs that change are not completely the

same. We think that it is better to show the full data of lipidomics experiments, rather than covering up any effect that may become of interest in the future by combining data.

What is the reason for using super-resolution in panel F, rather than regular epifluorescence microscopy? If the authors propose that without that they couldn't distinguish clusters from large droplets, it is worth pointing that out.

As suggested, we rewrote the sentence as follows: *“Finally, we performed super-resolution microscopy after labelling LDs with PLIN2 antibodies to differentiate between true large LDs and aggregated smaller LDs. This showed that the LD signal in the KO cells often corresponded to clustered small LDs, but individual large LDs were also observed (Figure 4F).”*

My main problem with this figure is that it does not go deeper. There are indeed more pre-LDs in the KO cells, but what does it mean? A reasonable interpretation is that this increase in pre-LDs is responsible for the increased TAG synthesis or the increased number of LDs. This could in principle be tested, e.g. if it is possible to reduce pre-LDs in the KO cells and show that TAG synthesis and LD number goes down. At a minimum, the authors should test whether they can rescue these phenotypes with the mutant constructs in Fig. 5 and determine if the same constructs rescue/fail to rescue both pre-LDs and TAG synthesis, for example.

We agree with the Reviewer and we think that indeed the increased number of pre-LDs is responsible for the increased TAG synthesis (see also Discussion). It would be very interesting to test if downregulation of components that regulate pre-LDs affect the phenotype caused by lack of spastin. However, we think that this goes beyond the scope of the current manuscript.

The analysis of lack of Spastin in Fig. 5 is intriguing and reveals a novel connection between Spastin, ER and LDs. However, mechanistic insight remains modest, other than showing that microtubule binding is not required to rescue ER collapse. The authors conclude that this has something to do with the ER shaping function of Spastin, but they have no positive evidence for that hypothesis (e.g., pre-LDs in the absence of TAG might lose or acquire certain general ER proteins, in a Spastin dependent manner, which then indirectly affects ER morphology).

As already mentioned in response to Reviewer 2, we realize that these data are preliminary, however we believe that they are worth reporting. We have stated in the Discussion that:

“Further experiments will be necessary to fully understand the cause of the ER collapse phenotype and the mechanistic role of spastin and other ER morphogens that interacts with spastin at the ER. In addition, the impact of autophagy on the ER phenotype remains to be investigated.”

The claim in the figure title "Increased LD biogenesis buffers lack of Spastin-M1 at the ER" is also not well supported. Yes, in the absence of all LD biogenesis, lack of Spastin-M1 somehow results in severe ER problems. However, it is not clear that

the increased LD biogenesis in Spastin mutants prevents ER collapse; for that, the authors would have to reduce LD biogenesis to levels in the wild type and test whether this reduction (rather than full abolishment) is sufficient to cause ER collapse. It seems a critical question to address is why lack of Spastin causes increased LD biogenesis, but no mechanism is proposed. The situation is further complicated by the fact that rescue is so dependent on the exact levels of M1 and M87. The clone in which protein levels appear to be closest to wild type (M1M87), confusingly, shows NO rescue, and rescue is best when versions of M1 and M87 lacking the microtubule binding domain are expressed about equally, far from the wild-type situation. This pattern does not easily lend itself to a mechanistic interpretation, and the authors do not offer any, beyond describing that absolute levels and ratios seem to be important. Finally, M1^{high}M87 and M1M87^{low} appear to be clones derived from the same transfection. Given the widely different expression levels and ER phenotypes, it strikes me as prudent to test multiple clones with the same expression levels to make sure that the observed phenotypes are indeed correlated with expression levels and not a unique property of a particular clone.

See answer to Reviewer 1 point 1.

Fig. 6 analyzes the role of Spastin in LD dispersion, and both section and figure title are overinterpreted. The figure title claims that Spastin mediates dispersion from the ER, but no analysis of the ER is presented in this figure. The section title claims that Spastin regulates dispersion, but there is no evidence of regulation; the observation is that in the absence of Spastin LDs dispersion is severely curtailed. One of the major conclusions of the section is that "lack of Spastin impairs the redistribution of LDs from the perinuclear region to the cell periphery upon glucose starvation but does not affect the speed of LDs that move." That nicely describes the phenotypes observed but does not explain why in the KO cells most LDs fail to move. Also, no data are provided as to the status of the microtubule cytoskeleton, other than that biochemically the fraction of non-tyrosinated tubulin seems normal. It would be important to know whether microtubule tracks are normal or altered (especially since the authors point out that non-tyrosinated microtubules are a particularly good substrate for Spastin). Those data might help rule out certain models. The authors show that the speed of LD motion in the KO cells is normal. Lack of dispersion could come about by droplets having a tendency to move towards instead of away from the clusters, at normal speeds. Have the authors examined more details of the motility?

We did not observe any visible difference in the organization of the MT cytoskeleton in KO cells. This of course does not rule out the possibility that the dynamic status of MTs is altered, but cannot be detected in fixed samples. Concerning the motility of LDs, it is technically challenging to further analyze these movies, because of the high number of LDs. Moving LDs move both away and towards the nucleus. The striking phenotype of spastin KO cells is the lack of motility of a large fraction of perinuclear LDs.

The authors also conclude that microtubule binding is necessary for LD dispersion. Here, their data is overinterpreted: they do not test whether wild-type M87 promotes dispersion, therefore lack of dispersion in the M87 mutant cannot be interpreted.

And wild-type M1highM87 fails to promote dispersion, so the absence of dispersion with mutant M1highM87 is also not unexpected. Given the complexity of expression levels/or M1-M87 ratios on phenotypes, it is conceivable that just the right ratio or levels of mutant M1 to mutant M87 would rescue.

We agree with the Reviewer that just the right ratio or levels of **wild-type** spastin-M1 would rescue the ER phenotype. However, our data clearly show that spastin-M1 lacking the MBD can rescue the ER phenotype when expressed at sufficient levels, indicating that spastin-M1 plays other MT-independent roles. It is true that we cannot exclude a role of spastin-M87 in the LD dispersion phenotype, and have therefore rewritten the corresponding paragraph. The two clones that rescue in fact do express some levels of spastin-M87.

Discussion, page 18: *“Our data do not allow to uniquely attribute to spastin-M1 this function, since even low amounts of spastin-M87, recruited to the ER by oligomerizing with spastin-M1, may be sufficient to restore LD movement.”*

We have also revised the Abstract and the last paragraph of the Introduction, and now mention a role of spastin, and not spastin-M1, in LD dispersion.

For comments on the rescue experiments and their interpretation and limitations, please also refer to the answer to Reviewer 1 point 1.

Finally, a minor point: the authors classify cells into clustered, intermediate or dispersed. It would be helpful to show examples of the intermediate state.

An example of the intermediate state can be seen in Figure 6A (labelled with i).

Fig. 7 summarizes lots of EM data that nicely show increased number and larger physical extents of LD/ER contacts in the KO cells. However, it is hard to know whether these increased contacts are a cause or consequence of the failed dispersion. In addition, in this case, they do not demonstrate that the phenotypes can be rescued by expression of wild-type Spastin.

We do not know if these increased contacts of ER/LD are cause of consequence of the failed dispersion. We think that this observation nicely reflects the perinuclear clustering, but unfortunately we have no additional mechanistic insights at this point.

April 7, 2020

RE: Life Science Alliance Manuscript #LSA-2020-00715-TR

Prof. Elena I Rugarli
CECAD Research Center
Institute for Genetics
Joseph-Stelzmann-Str. 26
Koeln 50931
Germany

Dear Dr. Rugarli,

Thank you for submitting your revised manuscript entitled "Microtubule-dependent and independent roles of spastin in lipid droplet dispersion and biogenesis". I appreciate the introduced changes and would be happy to publish your paper in Life Science Alliance pending final minor revisions necessary to meet our formatting guidelines:

- I would like to propose rephrasing the following sentence to prevent that readers think of PLIN2 as an autophagy substrate in the context of the autophagy inhibition experiments:

"Incubation of cells with HBSS and 3-methyladenine or bafilomycin A, which inhibit autophagy either at early or late steps, respectively, was however not sufficient to abrogate the increased levels of PLIN2 in KO compared to WT cells (Figure S4F, H)".

=> "Incubation of cells with HBSS and 3-methyladenine or bafilomycin A, which inhibit autophagy either at early or late steps, respectively, was however not sufficient to abrogate the increased levels of lipid droplets (visualized via PLIN2 levels) in KO compared to WT cells (Figure S4F, H)".

- Please upload figure S7 (figure S5 is currently uploaded twice)
- Please add the description to panel 7D in the figure legend
- Please add a scale bar to Figure 5A
- Boxed areas: boxes in Figure 2C are missing

A. FINAL FILES:

B. MANUSCRIPT ORGANIZATION AND FORMATTING:

Thank you for your attention to these final processing requirements.

Sincerely,

Andrea Leibfried, PhD
Executive Editor
Life Science Alliance
Meyerhofstr. 1

69117 Heidelberg, Germany
t +49 6221 8891 502
e a.leibfried@life-science-alliance.org
www.life-science-alliance.org

April 14, 2020

RE: Life Science Alliance Manuscript #LSA-2020-00715-TRR

Prof. Elena I Rugarli
CECAD Research Center
Institute for Genetics
Joseph-Stelzmann-Str. 26
Koeln 50931
Germany

Dear Dr. Rugarli,

Thank you for submitting your Research Article entitled "Microtubule-dependent and independent roles of spastin in lipid droplet dispersion and biogenesis". It is a pleasure to let you know that your manuscript is now accepted for publication in Life Science Alliance. Congratulations on this interesting work.

DISTRIBUTION OF MATERIALS:

Again, congratulations on a very nice paper. I hope you found the review process to be constructive and are pleased with how the manuscript was handled editorially. We look forward to future exciting submissions from your lab.

Sincerely,

Andrea Leibfried, PhD
Executive Editor
Life Science Alliance
Meyerohofstr. 1
69117 Heidelberg, Germany
t +49 6221 8891 502
e a.leibfried@life-science-alliance.org
www.life-science-alliance.org